# Associations of myeloid cells with cellular and humoral responses following vaccinations in patients with neuroimmunological diseases

Meng Wang [1,2,3], Adeline Dehlinger [1,2,3], Camila Fernández Zapata [1,2,3], Maya Golan[4], Gerardina Gallaccio [1,2,3], Leif E. Sander [5], Stephan Schlickeiser[6], Desiree Kunkel [7], Tanja Schmitz-Hübsch[1,2,3,8], Birgit Sawitzki[9], Arnon Karni [4,10], Julian Braun [11,12], Lucie Loyal [11,12], Andreas Thiel [11,12], Judith Bellmann-Strobl [1,2,3], Friedemann Paul[1,2,3,8,13,14], Lil Meyer-Arndt[8,11,13,14] & Chotima Böttcher [1,2,3,14] ✉

Disease-modifying therapies (DMTs) are widely used in neuroimmunological diseases such as multiple sclerosis (MS) and neuromyelitis optica spectrum disorder (NMOSD). Although these treatments are known to predispose patients to infections and affect their responses to vaccination, little is known about the impact of DMTs on the myeloid cell compartment. In this study, we use mass cytometry to examine DMT-associated changes in the innate immune system in untreated and treated patients with MS ($n = 39$) or NMOSD ($n = 23$). We also investigated the association between changes in myeloid cell phenotypes and longitudinal responsiveness to homologous primary, secondary, and tertiary SARS-CoV-2 mRNA vaccinations. Multiple DMT-associated myeloid cell clusters, in particular CD64$^+$HLADR$^{low}$ granulocytes, showed significant correlations with B and T cell responses induced by vaccination. Our findings suggest the potential role of myeloid cells in cellular and humoral responses following vaccination in DMT-treated patients with neuroimmunological diseases.

Neuromyelitis optica spectrum disorder (NMOSD) is a rare antibody-mediated autoimmune disorder of the central nervous system (CNS). Characteristic symptoms include acute attacks of optic neuritis, transverse myelitis and encephalopathic syndromes[1–3]. Due to some overlapping clinical features, it can easily be confused with multiple sclerosis (MS), and thus misdiagnoses are common.

While MS and NMOSD are both chronic demyelinating diseases, they exhibit distinct patterns of tissue injury and degeneration. In NMOSD, disease-specific auto-reactive IgG1 antibodies target the aquaporin-4 (AQP4) water channel protein on astrocytes leading to astrocytopathy and secondary demyelination and

neuron loss. Previous studies have investigated the unique immunophenotypes of NMOSD patients, which differ to some extent from those of MS patients[4–6]. However, there is a lack of research on innate immunophenotypes in patients with these neuroimmunological diseases, disregarding an integral aspect of (auto-) immunity. The relevance of the innate immune system is indicated, for instance, by elevated numbers of neutrophils infiltrating NMOSD lesions and in the cerebrospinal fluid during relapses[7–9]. Furthermore, neutrophils derived from NMOSD patients show a decreased functionality compared to neutrophils from MS patients or healthy individuals[10]. Our previous study

showed that there were only mild changes in the phenotype of immune cell populations in early MS[11].

Similar to MS patients, most NMOSD patients require early immunomodulatory treatment to prevent permanent neurological damage caused by disease relapses. While these disease-modifying therapies (DMTs) help reduce inflammation-related damage, they also predispose patients toward infections and decrease responsiveness to vaccinations, primarily due to the depletion or suppression of the lymphocyte compartment. Understanding vaccine responses in DMT-treated patients has become particularly relevant during the COVID-19 pandemic. However, the contribution of innate immune cells in this context remains poorly understood.

Myeloid cells, including circulating monocytes and neutrophils, take on an indispensable role in immune responses by stimulating naïve T cells and promoting their differentiation into effector cells, which in turn foster the survival and proliferation of B cells with the highest affinity to antigens[12–14].

Although detailed mechanisms of action of mRNA vaccines are still not fully understood, tissue-resident myeloid cells such as macrophages and DCs are known to play key roles at the site of injection. These cells facilitate the production of translated proteins that activate T and B cells, which are essential for mounting an effective immune response. Several studies have shown that the immunogenicity of COVID-19 vaccines varies among MS patients, with differences mainly dependent on the type of treatment they received at the time of vaccination. B cell depleting (BCD) therapies and sphingosine-1 phosphate (S1P) receptor antagonists had the greatest impact on the humoral (and in the case of S1P, cellular) vaccination response[15,16]. Importantly, studies investigating immune characteristics specific to MS in an inactivated state (I.e., without exposure to an antigen) have also concluded that immune cell compositions are more likely dependent on the type of treatment rather than the disease course[6]. However, detailed immune signature changes, particularly those in the innate immune compartment, upon primary and repeated exposure to a novel antigen in MS and NMOSD patients have not been evaluated. Furthermore, little is known about the effect of DMTs on the innate immune cell composition in NMOSD patients compared to MS patients.

In this study, we characterized changes in both the innate and adaptive arm of the immune system after DMTs in MS and NMOSD patients, compared with untreated patients. Changes in the myeloid compartment predominantly varied between diseases and types of treatments. Furthermore, we demonstrated the correlations of some DMT-associated myeloid cell sub-populations and humoral and T cell vaccination responses. Finally, we suggest that monitoring myeloid cell compartment alongside lymphocyte populations may aid physicians in better assessing a patient's immune status. However, the decision should be based on individual factors such as the specific DMT, the patient's neuroimmunological disease phenotype (e.g., disease subtype, duration, and severity), and their risk factors (e.g., aging, lifestyle, and comorbidities).

## Results

### Disease- and treatment-specific immunophenotypic signatures prior to primary antigen exposure

To investigate the effect of primary and follow-up exposure to a novel antigen on innate and adaptive immune signatures in patients with well-defined neuroimmunological diseases, including MS and the rare disease NMOSD, we examined a total of 62 patients (i.e., 39 with MS and 23 with NMOSD) before and after mRNA COVID-19 vaccination. Among them, 14 were untreated and 48 were on common immunomodulatory monotherapies (Supplementary Table 1). Medical histories, blood samples and nasopharyngeal swabs were collected prior to (baseline; T0) and 1 month after primary vaccination (i.e., 1st dose; T1), up to 6 months after secondary vaccination (i.e., 2nd dose; T2) as

well as up to 4 months after tertiary vaccination (i.e., 3rd dose; T3) (Supplementary Table 1 and Fig. 1A).

We first evaluated the effect of DMTs typically used in MS on innate and adaptive immune cell compositions before the first COVID-19 vaccination (T0). Using our CyTOF workflow as previously described[11] with some modifications (see Materials and Methods for more detail), whole blood samples were characterized using an antibody panel of 37 antibodies (i.e., *Panel A*, Supplementary Table 2). Clustering with the *FlowSOM* and *ConsensusClusterPlus* packages revealed a total of 18 immune cell clusters in all patients at all time points (Fig. 1B; see data pre-processing in Supplementary Fig. 1). Overall, HLA-DR⁻CD66b⁺CD16⁺ granulocytes (cluster 1, C1) were the most common cell type in the peripheral blood of all patients, regardless of their disease or therapy, representing 77.43% ± 7.62 of all CD45⁺ leucocytes (Fig. 1B). Before the primary vaccination (T0), the proportion of CD19⁺ cells (C15) was significantly reduced in MS patients treated with BCD therapies (i.e., anti-CD20 monoclonal antibodies (aCD20); ocrelizumab and rituximab) or the S1P receptor antagonist fingolimod (FTY) (Fig. 1C and D), as were natural killer (NK) cells (C10 for interferon β (IFNβ) and C16 for aCD20 and FTY), FcεR1aʰⁱCD11c⁺ DCs (C4), and CD14ᵈⁱᵐCD16ᵈⁱᵐ (C14) and CD4⁺ T cells (C6). Conversely, BCD and FTY therapy resulted in a higher proportion of granulocytes (C1 and C2, Fig. 1D) and CD14⁺CD16⁻ monocytes (C8 and C9) were increased in FTY- and IFNβ-treated MS patients (Fig. 1D).

We found that aCD20 therapies had a similar effect on reducing B cell counts in NMOSD patients (Fig. 1E, F) at T0. In contrast, NMOSD patients treated with the immunosuppressant mycophenolate mofetil (MMF) did not show a reduction in B cell abundance. In contrast to MS patients, the proportion of granulocytes (C1) in aCD20-treated NMOSD patients appeared to be comparable or lower than that in untreated patients. Similarly, MMF-treated NMOSD patients showed similar granulocyte abundance (C1) to that of untreated and aCD20-treated NMOSD patients (Fig. 1F). It is important to note that the results obtained from NMOSD patients in this study should be considered as observations rather than statistically significant findings, given the limited number of participants. Nevertheless, our results suggest that treatment-related changes in the granulocyte compartment of NMOSD patients differ from those detected in MS patients.

Next, we examined the changes in immune cell compositions across all four time points by using principal component analysis (PCA) to summarize the proportions of the 18 identified immune cell clusters in terms of the first two dimensions (Dim1 and Dim2) at T0, T1, T2 and T3 in both untreated and aCD20-treated MS and NMOSD patients (Fig. 2). The loading plots depict the cell populations that exhibited high variance and had a strong influence on a principal component (indicated by the length of the vector), highlighting differences between patients (i.e., between untreated and aCD20-treated patients), consistent with the results presented in Fig. 1. Granulocytes (C1), B cells (C15), DCs (C4), monocytes (C8, C9 & C14), NK cells (C10) and T cells (C6) significantly contributed to the first dimension (Dim1), accounting for 36.3% of the total variance-covariance, and thereby defining the overall variability, including those between untreated and aCD20-treated MS patients (Fig. 2A). Similar sets of clusters exhibited high variance among all treatment groups (i.e., untreated, aCD20-, FTY- and IFN-β-treated patients, Supplementary Fig. 2A). However, no changes in immune cell proportion were detected after vaccination (Supplementary Fig. 3), while treatment-driven differences were evident across all time points (Supplementary Figs. 2 and 4; Fig. 2C, E and G). In NMOSD patients, the granulocyte cluster C1 showed less influence on Dim1 (44% of the total variance-covariance) compared to other cell populations (Fig. 2B, untreated vs aCD20-treated NMOSD patients and Supplementary Fig. 2B, all treatment groups). Nevertheless, T cells (C6 and C18), DCs (C4) and monocytes (C7 and C9) also exhibited high variance and contributed

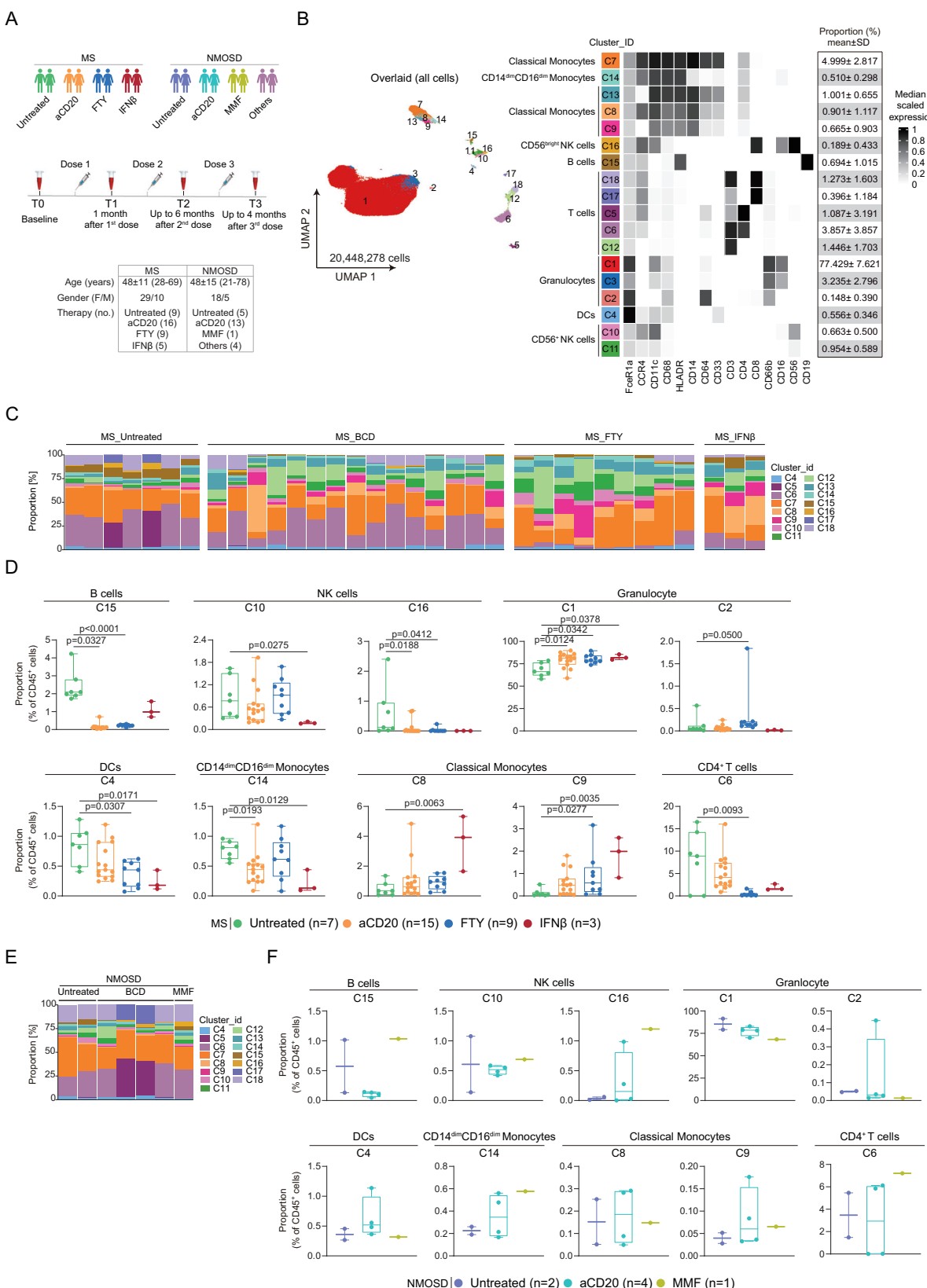

to the overall variability between conditions in NMOSD patients. Similar cell populations consistently showed substantial contributions to the overall variability between untreated and treated NMOSD patients across all four time points (Fig. 2B, D, F and H; Supplementary Fig. 2). These variabilities were not influenced by vaccination

(Supplementary Fig. 3), but predominantly varied between diseases and DMTs (Supplementary Fig. 4). Taken together, our findings suggest that the overall variability between samples, particularly between untreated and treated patients, was primarily driven by treatment rather than vaccination.

**Fig. 1 | Study design and immune cell characterization at the baseline (T0 prior to vaccination) using CyTOF (antibody panel A). A** Schematic overview of longitudinal study design, vaccine administration scheme and sample collection in relation to vaccinations across four time points for MS and NMOSD groups. Cohort information is shown in the bottom box. **B** UMAP projection, coloring indicates 1-18 clusters. Phenotypic heatmap of cluster identities depicting the median expression levels of selected markers per cluster. Heat colors of expression levels have been scaled for each marker individually (to the 1st and 5th quintiles) (black, high expression; white, no expression). **C** Proportion of each cluster (except granulocyte

clusters) from each sample in MS groups at T0. **D** Box plots of the ten differentially abundant clusters (mean ± SD) from untreated-MS (n = 7), aCD20-MS (n = 15), FTY-MS (n = 9), IFNβ-MS (n = 3) at T0. Each dot represents the value of each sample. Boxes extend from the 25th to 75th percentiles. Whisker plots show the min (smallest) and max (largest) values. The line in the box denotes the median. Kruskal-Wallis and Dunn's multiple comparison test. (E) Proportion of each cluster (except granulocyte clusters) from each sample in NMOSD groups at T0. (F) Box plots of the ten clusters as in **D** (mean ± SD) from untreated-NMOSD (n = 2), aCD20-NMOSD (n = 4), MMF-NMOSD (n = 1) at T0.

## In-depth characterization of myeloid and NK cell subpopulations and their potential link to humoral vaccination responses

Next, we investigated whether the changes in immune cell composition, which were attributed to DMTs and identified in Fig. 2, were associated with vaccine-specific antibody production (i.e., anti-SARS-CoV-2 spike protein subunit 1 (S1) immunoglobulin G (IgG)) following primary, secondary, and tertiary vaccination. In line with our previous research[16], we found that aCD20-treated MS and NMOSD patients, as well as FTY-treated MS patients, had no or very low levels of anti-S1 IgG at one month after the primary vaccination (T1) (Supplementary Fig. 5A). However, the number of S1 IgG-seropositive patients increased at T2 and T3 (Supplementary Fig. 5B, C). To evaluate the impact of lymphocyte depletion on the apparent dominance of myeloid cells and to comprehensively analyze changes in each major cell population more closely, we performed sub-clustering analyses of pre-gated CD19[+] B cells (antibody Panel B, Supplementary Table 3, Supplementary Fig. 6), CD66b[+] granulocytes, CD3[+] T cells (Supplementary Fig. 7), and other myeloid and NK cells (Supplementary Fig. 8).

Sub-clustering analysis of pre-gated CD19[+] B cells (Supplementary Fig. 1) revealed 18 distinct phenotypic clusters (Supplementary Fig. 6A, B). Most of the B cell sub-clusters were strongly depleted by aCD20 treatment (Supplementary Fig. 6C, D), except for CD20[-] sub-clusters B6 which remained unaffected. Similar changes in cluster abundance were observed in aCD20-treated NMOSD patients (Supplementary Fig. 6E). The proportion of CD20[+] B cell sub-clusters (B2, B3, B4, B5, B7, B9, B10, B12 and B16), which were affected by the respective treatment (as shown in Supplementary Fig. 6D), showed a positive correlation with anti-S1 IgG levels after vaccination (i.e., T1-T3, see Supplementary Fig. 6F). Therefore, the presence of these B cell subsets was most likely required for vaccine-specific IgG antibody production. Additionally, we identified one CD20[-] and six CD20[+] B cell subsets (CD20[-]: B6, a mixed population of CD20[-]IgM[-]IgD[low]CD138[+] CD38[+] CD27[+]Ki67[+]IgGK[+] and IgA[+] B cells; CD20[+]: B1, CD20[+]IgM[hi]IgD[+]; B8, CD20[+]IgA[+]; B13, CD20[+]Tbet[hi]CD11c[+]; B15, CD20[+]CD24[hi]CD38[hi]; B17, CD20[+]IgM[+]IgD[hi]; B18, CD20[lo]IgM[lo]IgD[-]CD38[hi]HLADR[hi]CD1c[+]). The proportions of these subsets were not significantly affected by DMT but showed a positive correlation with antibody production (Supplementary Fig. 6F).

CD66b[+] granulocytes were sub-clustered into 18 sub-clusters (Supplementary Figs. 1 and 7; Fig. 3A, B). Our results demonstrated treatment-dependent increase in the proportion of two granulocyte sub-clusters (G1: CCR4[hi]CXCR4[low]HLA-DR[-] and G3: HLA-DR[-]CCR4[+]CXCR4[low]) as well as a reduction of one cluster, G6: CD14[+], in MS patients treated with aCD20 at T0 (Fig. 3C; see Supplementary Fig. 9 for non-significantly differential abundant clusters). No differences were found in the proportions of all three sub-clusters between FTY- and IFN-β-treated patients and untreated patients. However, the same changes were not evident in aCD20-treated NMOSD patients (Fig. 3D). On the other hand, an NMOSD patient treated with the immunosuppressants MMF showed a strong reduction in the proportion of G1 (HLA-DR[-]CD16[-]CCR4[hi]CXCR4[low]), G3 (HLA-DR[-]CD16[+]CCR4[hi]CXCR4[low]) and G6 (HLA-DR[-]CD14[+]) at T0 (Fig. 3D). Nonetheless, as mentioned above, the results obtained from NMOSD patients should be interpreted with caution due to low statistical power of testing.

Among these DMT-affected granulocyte sub-clusters, HLA-DR[-]G1 and G9 exhibited a strong contribution to the overall variability in all

MS patients (including DMT-mediated differences) across all post-vaccination time points (i.e., T1-T3) as shown in the PCA (Fig. 3E). Interestingly, in aCD20-treated MS patients, the proportions of both G1 and G9 were negatively correlated with anti-S1 IgG antibody production (Fig. 3F (upper panel)). Moreover, we detected significant correlations between anti-S1 IgG levels and other clusters with high variance in the PCA, i.e., HLA-DR[-]CCR4[-] G11, G13 and G17 (negative correlation, Fig. 3F, upper panel) as well as HLA-DR[low/dim] G7, G10 and G15 (positive correlation, Fig. 3F, lower panel). However, we could not detect any significant correlations between granulocyte sub-clusters and anti-S1 IgG antibody production in untreated and FTY-treated MS patients.

Unlike MS patients, untreated NMOSD patients showed a significant correlation between anti-S1 IgG antibody production and the proportion of CD14[+] G6 (positive correlation) and HLA-DR[low/dim] G18 (negative correlation) (Fig. 3G). Furthermore, we identified similar granulocyte populations which highly contributed to the overall variability in both untreated and treated NMOSD patients (i.e., aCD20 and MMF; Fig. 3H). Consistent with the findings in MS patients, we observed a positive correlation between HLA-DR[dim/+] sub-clusters (G2 and G7, but not G10) and anti-S1 IgG levels in aCD20-treated NMOSD patients (Fig. 3I). However, in contrast to untreated MS patients, the proportion of CD64[+] sub-clusters G12 and G16 negatively correlated with anti-S1 IgG production after vaccination (Fig. 3I). No significant correlation between granulocyte sub-clusters and antibody production in the MMF-treated NMOSD patient was detected, possibly due to low statistical power. In summary, the granulocyte sub-clusters that showed a positive correlation with anti-S1 IgG levels were predominantly HLA-DR[low/+] (Supplementary Fig. 10), while those with a negative correlation commonly exhibited higher CXCR4 expression levels (Supplementary Fig. 10).

Pre-gating and sub-clustering analysis of other myeloid and NK cells (MNK) using antibody panel A (Supplementary Fig. 8) resulted in three NK cell sub-clusters (N5, N6, and N7) and fifteen sub-clusters of myeloid cells (Fig. 4A, B). Prior to vaccinations (at T0), we detected DMT-mediated increase (Fig. 4C) or decrease (Fig. 4D) in the proportion of eleven myeloid cell sub-clusters, as well as a decrease in one NK cell sub-cluster (N6) (Fig. 4E) in MS patients. However, these changes in cell compositions appeared to differ in NMOSD patients, apart from M13 (Fig. 4F–H). In aCD20-treated MS patients, we observed positive correlations between anti-S1 IgG antibody production and three myeloid cell sub-clusters (M11: CD14[-]CD16[-]CD64[+]; M15: CD14[lo]CD16[+]; M17: CD14[+]CD16[+]CD68[hi]), as well as one NK cell sub-cluster (N7: CD56[+]CD16[+]CD8[hi]) (Fig. 4I). Additionally, we detected a negative correlation between the CD14[-]CD16[-] monocyte sub-cluster (M10) and anti-S1 IgG levels (Fig. 4I). However, there was no significant correlation between MNK cell proportions and anti-S1 IgG levels in untreated, FTY-, or IFN-β-treated MS patients, as well as in untreated or MMF-treated NMOSD patients. In contrast to aCD20-treated MS patients, aCD20-treated NMOSD patients showed a positive correlation between M10 and a negative correlation between M17 and anti-S1 IgG production (Fig. 4J). Additionally, the CD8[-] NK cell sub-cluster (N6) was negatively correlated with anti-S1 IgG antibody levels (Fig. 4J).

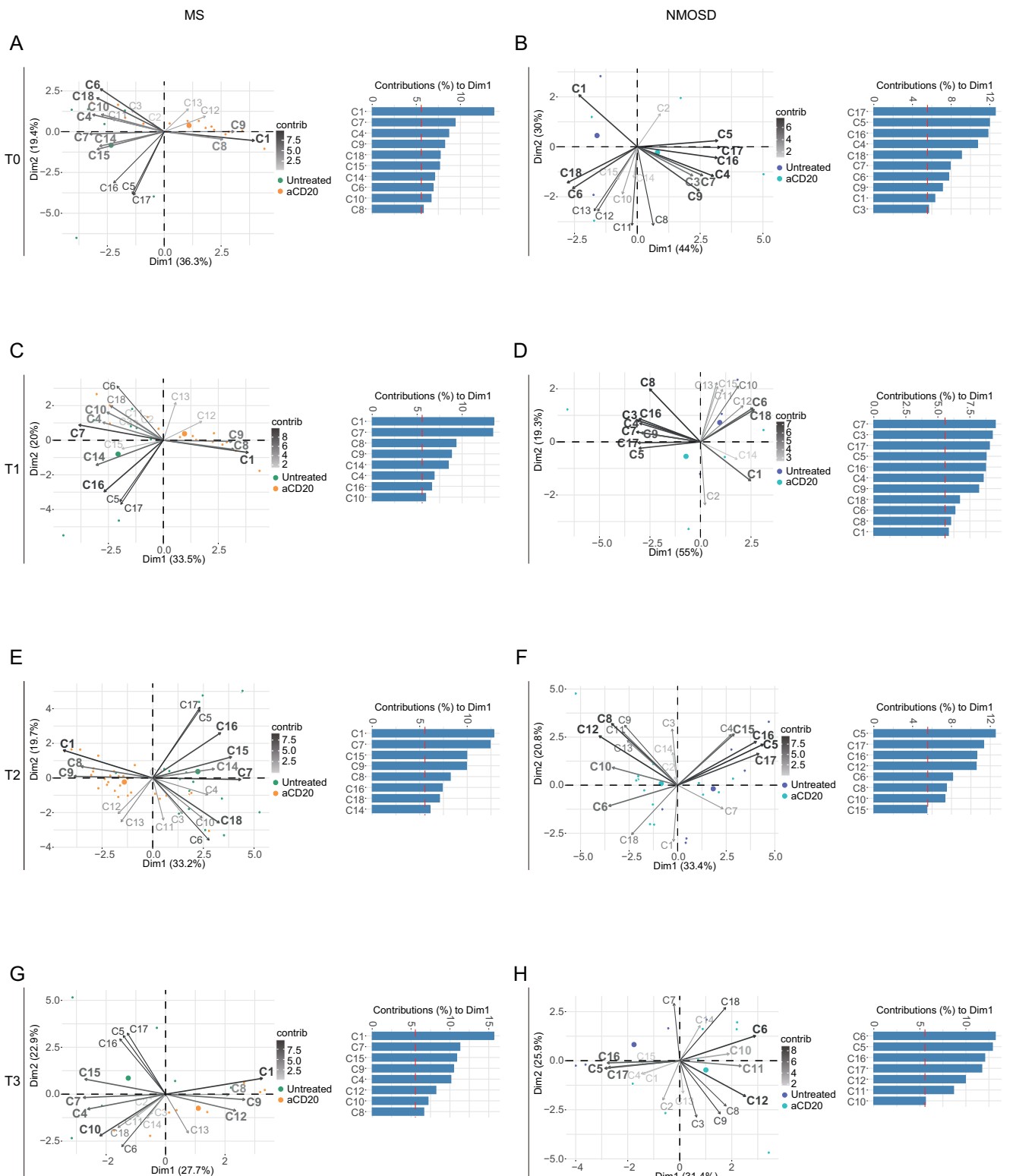

**Fig. 2 | Contribution of immune cell sub-populations to aCD20-mediated differences in cell composition at different timepoints depicted by principal components analysis (PCA). A**–**H** PCA for the 18 identified immune clusters in untreated and aCD20-treated MS **A**, **C**, **E** & **G** and NMOSD patients **B**, **D**, **F** & **H** at T0 **A**, **B**, T1 **C**, **D**, T2 **E**, **F** and T3 **G**, **H**. Each point represents one sample's scores on the first 2 dimensions (Dim1 and Dim2). Each vector (arrow) shows the loadings of each cell cluster on the first 2 principal components. Gray scale indicates the contribution value of variables to Dim1 and Dim2. The red dashed line on the graph indicates the expected average contribution. The graph shows top variables (with a contribution larger than average) contributing to Dim1.

## Link between myeloid cell involvement and SARS-CoV-2-specific CD4+ T cell responses

Next, we assessed the heterogeneity and composition changes of T cell populations in untreated and treated patients after vaccination (i.e.,

T1-T3). Using antibody panel A, we identified 18 T cell sub-clusters (Fig. 5A, B; Supplementary Fig. 7). The proportion of CD4+CCR7+ (T3) sub-cluster was significantly reduced in FTY-treated MS patients, while the proportion of CD4-CD8-CCR7- double negative (DN) sub-clusters

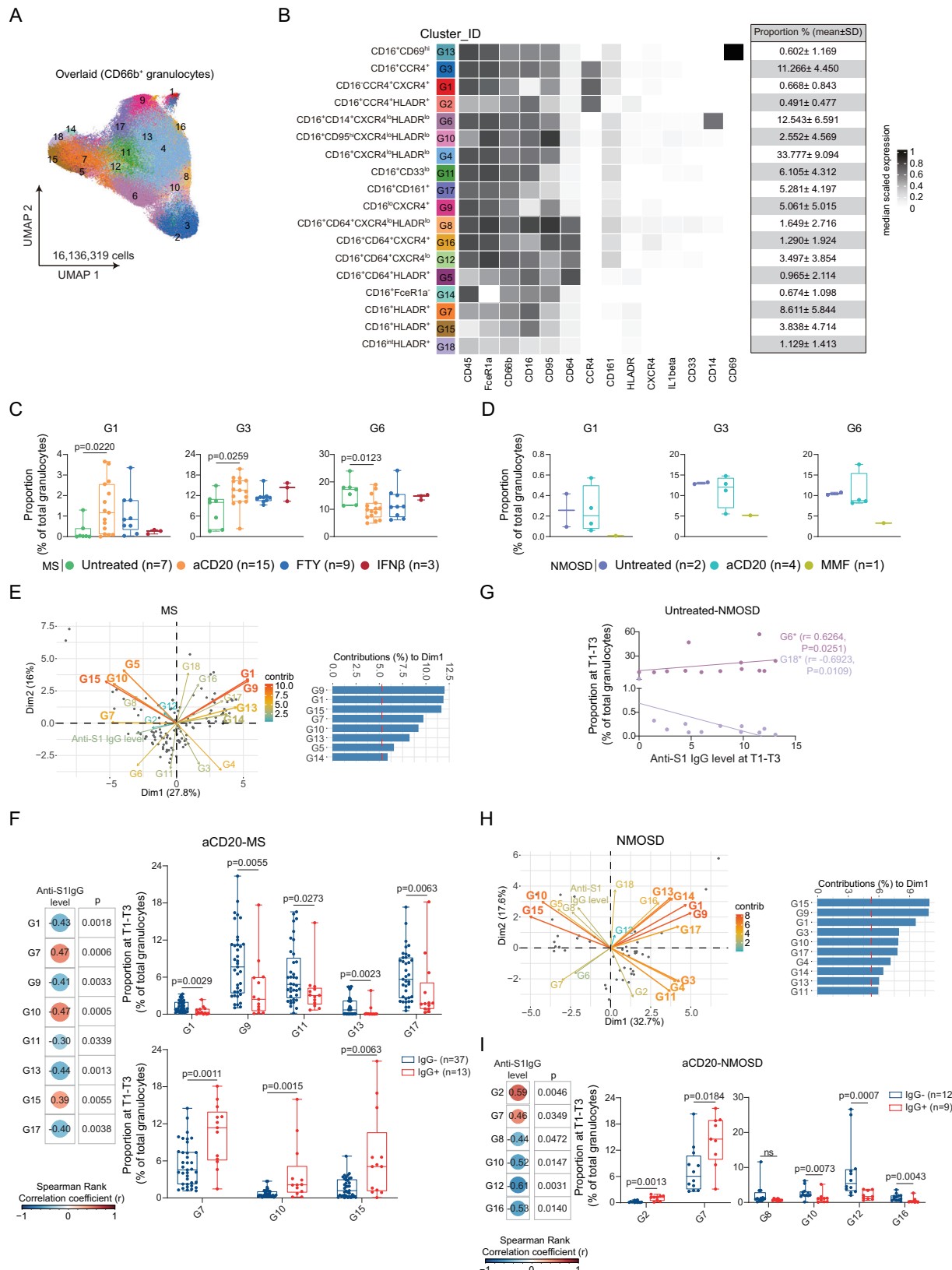

(T7 and T9) was found to be increased (Fig. 5C). Additionally, we detected an increased proportion of two DN sub-clusters (T7 and T8) and a decreased proportion of CD8+HLADR^hi T cells (T12) in MS patients on BCD therapies (Fig. 5C). We did not observe any significant differences in T cell sub-clusters between untreated, aCD20- and MMF-treated NMOSD patients (Fig. 5D). To investigate whether these

changes in T cell composition were related to T cell vaccination responses, we examined antigen-specific CD4+ T cell reactivity to SARS-CoV-2-spike glycoprotein, using the S-I peptide pool post-vaccination. Untreated MS patients and those on BCD therapies displayed an increased S-I-specific CD40L+4-1BB+ CD4+ T cell response after vaccination, irrespective of their neuroimmunological disease

**Fig. 3 | Association of granulocyte sub-population with anti-S1 IgG antibody production at T1-T3 in MS and NMOSD groups. A** UMAP plots of granulocytes. **B** Phenotypic heatmap depicting the median marker expression levels of each granulocyte clusters. **C** Proportion of the three differentially abundant clusters (mean ± SD, Kruskal-Wallis and Dunn's multiple comparison test), compared between Untreated-MS ($n = 7$), aCD20-MS ($n = 15$), FTY-MS ($n = 9$), IFNβ-MS ($n = 3$) at T0. **D** Proportion of the three clusters as in **C** (mean ± SD) between Untreated-NMOSD ($n = 2$), aCD20-NMOSD ($n = 4$), MMF-NMOSD ($n = 1$) at T0. **E** PCA for all 18 granulocyte clusters and IgG level in MS groups. Each point represents one sample's scores ( = one patient) on the first 2 dimensions (Dim1 and Dim2). Each arrow shows the loadings on the first 2 principal components. The graph shows top variables (with a contribution larger than average) contributing to Dim1. **F** Heatmap of the Spearman correlation coefficients between the proportion of clusters and antibody levels in aCD20-MS group (Nonparametric Spearman correlation test (r), two-sided). Box plots (right panel) showing correlated cluster proportion compared between IgG⁻ ($n = 37$) and IgG⁺ ($n = 13$) groups in aCD20-MS group (Kruskal-Wallis and Dunn's multiple comparison test). **G** Scatter plots showing correlation between the proportion of clusters and antibody levels in Untreated-NMOSD groups (Nonparametric Spearman correlation test (r), two-sided). **H** PCA for the 18 granulocyte clusters and IgG level in NMOSD groups (as in **E**). The graph shows top variables (with a contribution larger than average) contributing to Dim1. **I** Heatmap of the correlation between the proportion of clusters and antibody levels in aCD20-NMOSD group (as in **F**, Nonparametric Spearman correlation test (r), two-sided). Box plots showing correlated cluster proportion compared between IgG⁻ ($n = 12$) and IgG⁺ ($n = 9$) groups in aCD20-MS group (Kruskal-Wallis and Dunn's multiple comparison test). Each dot represents the value of each sample. For all Box plots, boxes extend from the 25th to 75th percentiles. Whisker plots show the min (smallest) and max (largest) values. The line in the box denotes the median. The n numbers (n) are defined as biologically independent samples.

(Fig. 5E, F). However, no significant increase of S-I-specific CD4⁺ T cell reactivity was detected in FTY-treated MS patients, which is consistent with our previous findings[15]. Correlation analysis revealed a negative correlation between S-I-specific CD4⁺ T cells (depicted as Stim.Index) and three CD4⁺/CD8⁺ICOS^loCD226⁺ (T2, T10 and T14) T cell sub-clusters in untreated MS (Fig. 5G). In addition, we observed a negative correlation between S-I-specific CD4⁺ T cells and the CD4⁺HLADR^hiICOS⁺CD226⁺ sub-cluster (T1) in aCD20-treated MS patients (Fig. 5H). However, in FTY-treated MS patients, two CD4⁺ICOS^loCD226^-/lo T cell sub-clusters (T3 and T5) were positively correlated with S-I-specific CD4⁺ T cell activity (Fig. 5I). In untreated NMOSD patients, in contrast to untreated-MS patients, we found a positive correlation between four CD4⁺/CD8⁺ICOS⁺CD226⁺ T cell sub-clusters (T1, T4, T6 and T15) and S-I-specific CD4⁺ T cell activity (Fig. 5J). Only a positive correlation between S-I-specific CD4⁺ T cells and the CD8⁺HLADR^hi sub-cluster (T12) was detected in aCD20-treated NMOSD patients (Fig. 5K).

Given the links identified between granulocytes as well as other myeloid and NK cells and the humoral vaccination responses, we further investigated whether immunophenotypic changes of these cell compartments correlated with vaccine-induced spike-specific CD4⁺ T cell responses. Interestingly, the correlations observed in MS patients were different from those in NMOSD patients (Fig. 6A). Specifically, in untreated MS patients, we found a positive correlation between HLA-DR^low G7 sub-cluster and both vaccine-specific antibody production and S-I-specific CD4⁺ T cell reactivity (Fig. 6B). However, in aCD20-treated MS patients, the G7 sub-cluster showed a negative correlation with vaccine-specific CD4⁺ T cells (Fig. 6C). Furthermore, the CXCR4^low sub-clusters (G3 and G13), which were negatively correlated with anti-S1 IgG antibody production, were also negatively linked to S-I-specific CD4⁺ T cell responses in untreated and aCD20-treated MS patients (Fig. 6B). In FTY-treated MS patients, we observed a positive correlation between the CD16^loCXCR4^lo sub-cluster (G17) and S-I-specific CD4⁺ T cell reactivity (Fig. 6D). In untreated NMOSD patients, in contrast to untreated MS patients, the CXCR4^low sub-cluster (G11) was positively associated with S-I-specific CD4⁺ T cell reactivity (Fig. 6E). However, in aCD20-treated NMOSD patients, the CD64⁺CXCR4^lo sub-cluster G12, which was negatively correlated with anti-S1 IgG antibody levels (Fig. 3I), exhibited a positive correlation with S-I-specific CD4⁺ T cell responses (Fig. 6F).

In line with the findings on granulocyte sub-clusters, we also found correlations between myeloid (M) and NK (N) cell sub-clusters and S-I-specific CD4⁺ T cells (Fig. 7A–E). Specifically, we detected positive correlations between the NK cell sub-clusters N5, N6 and N7, and the monocyte cell sub-cluster M17 in untreated MS patients, while slightly inverse correlations were observed in aCD20-treated MS patients (Fig. 7B, C). In MS patients treated with FTY, the proportion of HLADR⁺CCR7⁺CD14⁻CD16⁻ (M13) myeloid cells showed a positive association with S-I-specific CD4⁺ T cell responses (Fig. 7D).

In contrast to aCD20-treated MS patients, in untreated NMOSD patients, the proportion of two HLADR⁺CCR7⁺CD14⁻CD16⁻ myeloid cell clusters (M16 and M18) negatively correlated with CD4⁺ T cell reactivity (Fig. 7E).

## Discussion

In this study, we conducted in-depth profiling of immunophenotypic signatures to identify distinct immune cell (sub-)populations in MS and NMOSD patients undergoing different immunomodulatory treatments. We then investigated these subpopulations and their link to cellular and humoral immune responses in the context of primary and booster SARS-CoV-2 vaccinations, serving as correlates of exposure to novel antigens. It is well-known that immunomodulatory therapies can hamper vaccination responses and increase risk of infection, despite their high efficacy in managing disease activity[17–22]. Our results demonstrate differential changes in the distribution of sub-populations of both the lymphocyte and myeloid cell compartment in MS and NMOSD patients treated with DMTs. These changes included an increased proportion of granulocytes and changes in the distribution of sub-populations in this cell type. Furthermore, some of these sub-populations showed either positive or negative correlations with both cellular and humoral responses after vaccination. Our findings suggest that, in addition to lymphocytes, the distribution of innate immune cell sub-populations is also altered by DMTs. These alterations may consequently affect the risk of infection and/or responses to vaccination. Antigens/pathogens stimulate cellular and humoral immune responses through similar mechanisms irrespective of the specific infection or vaccine. Thus, studying immune responses to a single antigen/pathogen can provide valuable insights into broader immune responses to other antigens/pathogens, such as infections and vaccines.

Innate immune cells, particularly neutrophil granulocytes, are the first line of defense of the immune system with diverse functions, including atypical antigen presentation as well as initiation and regulation of adaptive immunity[23–30]. For example, aged neutrophils were characterized by upregulation of C-X-C motif chemokine receptor type 4 (CXCR4), a migration and homing factor, in mouse models[24,31,32] and exhibited a higher phagocytic activity compared to non-aged neutrophils[24]. This distinct behavior of "experienced" aged neutrophils during inflammation may allow them to instantly translate inflammatory signals into immune responses. CD64 expression, which serves as receptor for the Fc region of IgG and can increase over 10-fold in activated neutrophils compared to resting ones, has long been used as a surrogate marker for infectious disease and inflammatory processces[33–35]. However, little is known about the CD64⁺ neutrophils in patients with MS or NMOSD and how they are associated with DMTs. In our analyses, we characterized heterogeneity of granulocytes and detected proportional changes of various sub-populations in response to DMTs, including CXCR4^dim/lowHLA-DR⁻, HLA-DR^low/⁺, and CD64⁺

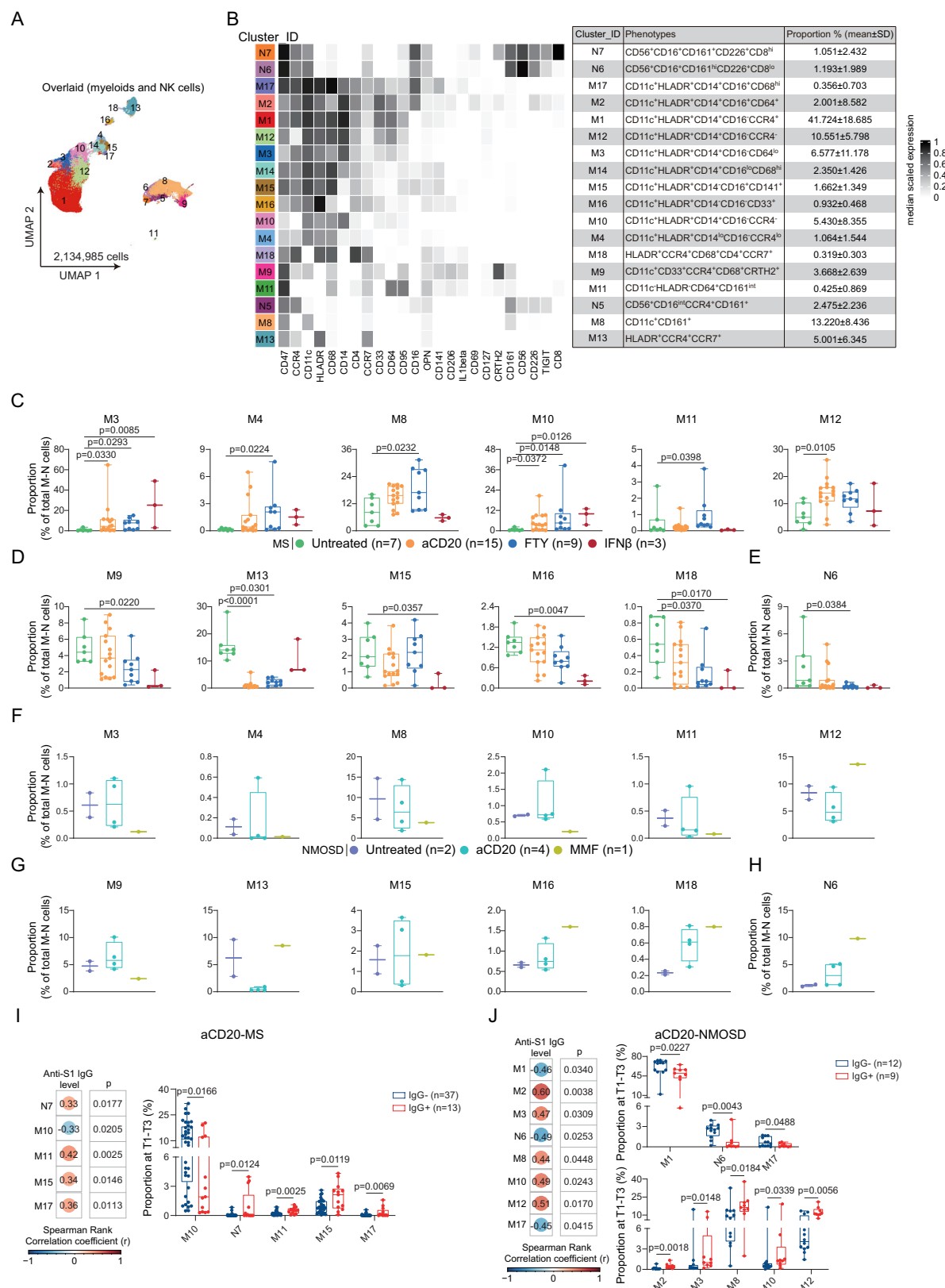

granulocytes. Most of them were negatively correlated with humoral vaccination responses in aCD20-treated MS and NMOSD patients as well as untreated patients, possibly due to their intercorrelation with the B cell proportion. Furthermore, CD64⁺ granulocytes were positively (i.e., in aCD20-NMOSD patients) linked to S-I-specific CD4⁺ T cell reactivity.

Interestingly, all granulocyte sub-clusters, which positively correlated with anti-S1 IgG levels in aCD20-treated MS and NMOSD patients, were characterized as HLA-DR^low/+ granulocytes, suggesting APC-like functions. Consistently, one of these sub-clusters (i.e., G7) was also positively linked to S-I-specific CD4⁺ T cell reactivity. However, sub-cluster G7 was negatively correlated with CD4⁺ T cell activation in

**Fig. 4 | Association of myeloid and NK cell sub-population with anti-S1 IgG antibody production at T1-T3 in MS and NMOSD groups. A** UMAP plots colored by cluster ID for 1–18 clusters of MNK cell determined using the FlowSOM algorithm. **B** Phenotypic heatmap depicting the median expression levels of selected markers per MNK cell cluster as defined in the table. **C**–**E** Proportion of the twelve differentially abundant clusters (mean ± SD) between Untreated-MS (*n* = 7), aCD20-MS (*n* = 15), FTY-MS (*n* = 9), IFNβ-MS (*n* = 3) at T0. Each dot represents the value of each sample. Boxes extend from the 25th to 75th percentiles. Whisker plots show the min (smallest) and max (largest) values. The line in the box denotes the median. Kruskal-Wallis and Dunn's multiple comparison test. **F**–**H** Proportion of the twelve clusters as in **C**–**E** (mean ± SD) between Untreated-NMOSD (*n* = 2), aCD20-NMOSD

(*n* = 4), MMF-NMOSD (*n* = 1) at T0. **I, J** Heatmap of the Spearman correlation coefficients between the proportion of MNK cell clusters and antibody levels at T1-T3 in aCD20-MS group **I** and aCD20-NMOSD group **J**. Nonparametric Spearman correlation test (r), two-sided. Box plots showing the proportion of correlated clusters compared between IgG⁻ and IgG⁺ groups at T1-T3 in aCD20-MS group **I** (IgG⁻, *n* = 37; IgG⁺, *n* = 13) and aCD20-NMOSD group **J** (IgG⁻, *n* = 12; IgG⁺, *n* = 9). Each dot represents the value of each sample. Boxes extend from the 25th to 75th percentiles. Whisker plots show the min (smallest) and max (largest) values. The line in the box denotes the median. Kruskal-Wallis and Dunn's multiple comparison test. All n numbers are defined as biologically independent samples.

aCD20-treated MS patients, again suggesting disease- or treatment-mediated differences in granulocytes functions, which are in line with previously published study on neutrophil alteration in NMOSD patients[7–10]. Due to the limited number of NMOSD patients, we were unable to detect any significant differences in the heterogeneity of granulocytes in these diseases. Nevertheless, our findings suggest that potential neutrophils may be involved in mounting an antigen-specific immune response upon primary and subsequent antigen exposure, which can be modulated by DMTs. They may play roles in infection and/or vaccine-specific humoral and cellular responses through their phagocytic and/or atypical APC activities, although the latter is still a matter of debate.

Similar to changes in granulocytes, we have detected DMT-mediated changes in the proportion of myeloid cell sub-clusters, some of which were significantly correlated with both anti-S1 IgG levels and S-I-specific CD4⁺ T cell reactivity. In untreated NMOSD patients, two CD127low myeloid sub-clusters showed opposite associations with S-I-specific CD4⁺ T cell reactivity (one sub-cluster had a positive association, while the other sub-cluster had a negative association). Interestingly, the associations of these sub-clusters with anti-S1 IgG levels were reversed in aCD20-treated NMOSD patients. CD127⁺ monocyte subsets are known to retain a hypo-inflammatory phenotype during highly inflammatory conditions as shown in COVID-19 and rheumatoid arthritis[36]. A strongly positive correlation between CD14⁺CD127⁺ myeloid cells (M2, M12) and anti-S1 IgG levels was only found in aCD20-treated NMOSD patients. In untreated NMOSD patients, M12 was negatively associated with S-I-specific CD4⁺ T cell reactivity. Again, these findings demonstrate differential cellular responses in different disease and treatment conditions.

In addition to granulocytes and monocytes, we also detected significantly negative correlations between the proportion of the CD8lowCD56⁺CD161⁺ (N6) NK sub-population and anti-S1 IgG antibody production in aCD20-treated NMOSD patients as well as a negative correlation with the S-I-specific CD4⁺ T cell response in aCD20-treated MS patients. However, this sub-population showed a positive association with CD4⁺ T cell responses in untreated MS patients. We furthermore detected another small population of NK cells, i.e., the N7 sub-cluster, which was associated with SI-specific CD4⁺ T cell responses in untreated and aCD20-treated MS patients. This NK cell population expressed CD8, CD95, CD69, CD161, and the chemoattractant receptor CRTH2, which has been implicated in various inflammatory diseases and is thought to be involved in immune cell recruitment and activation[37,38]. Again, N7 was positively correlated with S-I-specific CD4⁺ T cell reactivity in untreated MS patients but negatively in aCD20-treated MS patients. This could indicate that downstream effects of B cell depletion may indirectly affect the function of NK cells or their interaction during an immune response.

Limitations of our study include the different group sizes and dropouts (i.e., missing data), which may affect statistical testing to compare study outcomes between groups. However, our main groups for analysis, untreated and aCD20/FTY-treated MS patients, are similarly sized, enabling comparison between treatment-naïve and treated

patients. Although the limited number of untreated and aCD20-treated NMOSD patients may impact the statistical power to detect subtle differences, our findings could serve as a basis for generating hypothesis for larger-scale studies that can offer more definitive conclusions within a broader NMOSD population. Additionally, our study has yielded valuable insights into the immune response patterns of innate immune cells in the context of DMTs and vaccination. Understanding these patterns can contribute to our understanding of how these innate immune cell populations may respond to (novel) infections and vaccines, extending beyond the COVID-19 vaccine and patients with neuroimmunological diseases. Therefore, investigating the changes in immune phenotypes of these innate immune cells, particularly granulocytes, before and after vaccination, could provide a framework for assessing immune responses in different scenarios. Another limitation is the absence of long-term follow-up data on infection rates and infection severity. Consequently, drawing conclusions regarding the clinical impact of the observed immune phenotypic changes is constrained. Finally, it is important to note that multiple other factors can affect both vaccination responses and the immune cell status, such as comorbidities. However, none of the participants in this study had comorbidities related to the immune system.

In conclusion, our study demonstrates the potential impact of lymphocyte-targeted DMTs, such as aCD20 antibodies, on the innate immune system. These effects can either support or inhibit B cell and T cell reactivity, thereby influencing the response to (novel) antigens. Importantly, these effects are also modulated by the underlying disease. Therefore, we suggest considering the assessment of innate immune status in the clinical monitoring of neuroimmunological patients to evaluate treatment efficacy or make decisions regarding therapy and vaccination. However, such decisions should be tailored to individual factors, such as treatment type, clinical and immunological patient characteristics, and risk factors that could influence a patient's susceptibility to adverse outcomes, treatment response, or vaccine effectiveness. It is essential to note that determining the ultimate benefit of this monitoring requires large-scale clinical studies. Additionally, further investigations, such as in vitro experiments with isolated human immune cells or in vivo experiments using animal model, are necessary to elucidate the roles of NK cells, granulocytes, and other myeloid cells in humoral and cellular responses to repeated antigen exposure or vaccination.

## Methods
The data analyzed here was collected as part of the CCC study at the Charité−Universitätsmedizin Berlin, which was approved by the ethics committee of the Charité−Universitätsmedizin Berlin (Ethikkommission der Charité−Universitätsmedizin Berlin; registration number EA2/224/21) in accordance with the Declaration of Helsinki of 1964 and its later amendments. Patients were recruited via the Charité's MS outpatient clinic. All study participants provided informed consent before any study-related procedures were undertaken and did not receive compensation.

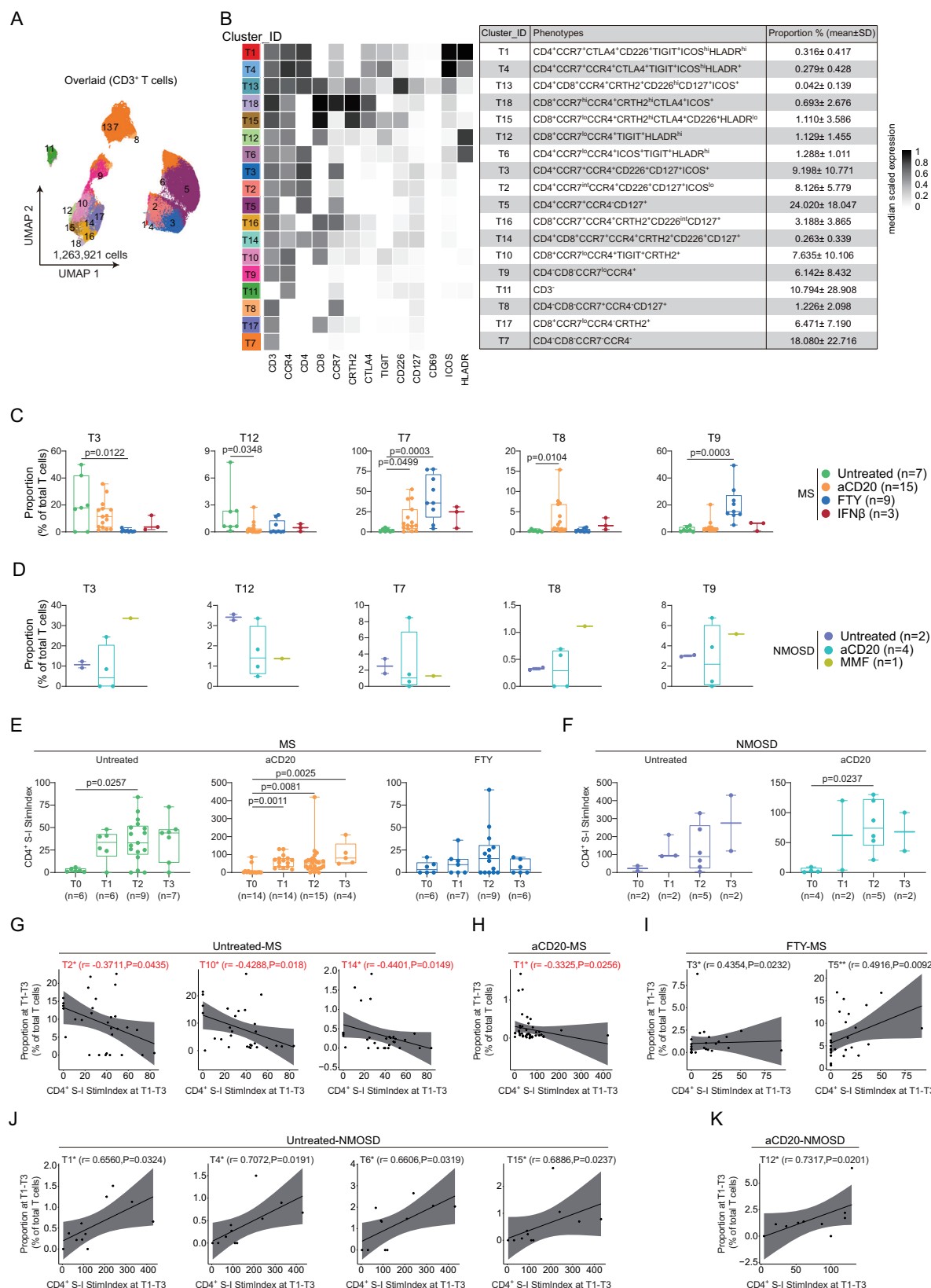

## Study design

To investigate the effect of primary and follow-up exposure to a novel antigen on innate and adaptive immune signature in patients with well-defined neuro-immunological diseases including NMOSD and MS, we recruited a total of 62 patients, including untreated patients and patients on distinct immunomodulatory treatments (i.e., 39 MS and 23

NMOSD) before and after mRNA COVID-19 vaccinations. Please see Supplementary Table 1 for clinical and demographic characteristics. The inclusion criteria were: (1) MS diagnosis according to the McDonald criteria of 2017 or NMOSD diagnosis according to Wingerchuk (2015), (2) stable disease for at least 3 months (no acute relapse therapy, no clinical progression or new symptoms suggestive of relapse, no

**Fig. 5 | Changes in CD3⁺ T cell composition and the correlation between S-I-specific CD4⁺ T cell reactivity and defined sub-populations at T1-T3 in MS and NMOSD patients. A** UMAP plots colored by cluster ID for 1–18 clusters of T cell determined using the FlowSOM algorithm. **B** Phenotypic heatmap depicting the median expression levels of selected markers per T cell cluster as defined in the table. **C** Proportion of the five differentially abundant clusters (mean ± SD) between Untreated-MS ($n = 7$), aCD20-MS ($n = 15$), FTY-MS ($n = 9$), IFNβ-MS ($n = 3$) at T0. Each dot represents the value of each sample. Boxes extend from the 25th to 75th percentiles. Whisker plots show the min (smallest) and max (largest) values. The line in the box denotes the median. Kruskal-Wallis and Dunn's multiple comparison test. **D** Proportion of the five clusters as in **C** (mean ± SD) between Untreated-NMOSD ($n = 2$), aCD20-NMOSD ($n = 4$), MMF-NMOSD ($n = 1$) at T0. **E, F** The CD4⁺ T cell reactivity between four timepoints (T0-T3) in different

MS groups **E** (Untreated-MS group, $n = 6, 6, 9$ and $7$, sequentially; aCD20-MS group, $n = 14, 14, 15$ and $4$, sequentially; FTY-MS group, $n = 6, 7, 9$ and $6$, sequentially) and NMOSD groups **F** (Untreated-NMOSD group, n = 2, 2, 5 and 2, sequentially; aCD20-NMOSD group, $n = 4, 2, 5$ and $2$, sequentially). Each dot represents the value of each sample. Boxes extend from the 25th to 75th percentiles. Whisker plots show the min (smallest) and max (largest) values. The line in the box denotes the median. Kruskal-Wallis and Dunn's multiple comparison test. **G–K** Scatter plots showing correlation between the proportion of T cell cluster and CD4⁺ S-I Stim.Index at T1-T3 in Untreated-MS group **G**, aCD20-MS group **H**, FTY-MS group **I**, Untreated-NMOSD group **J**, and aCD20-NMOSD group **K**. The red text denotes the negative correlation. Nonparametric Spearman correlation test (r), two-sided. Black lines and gray shadows represent the best-fitted smooth line and 95% confidence interval.

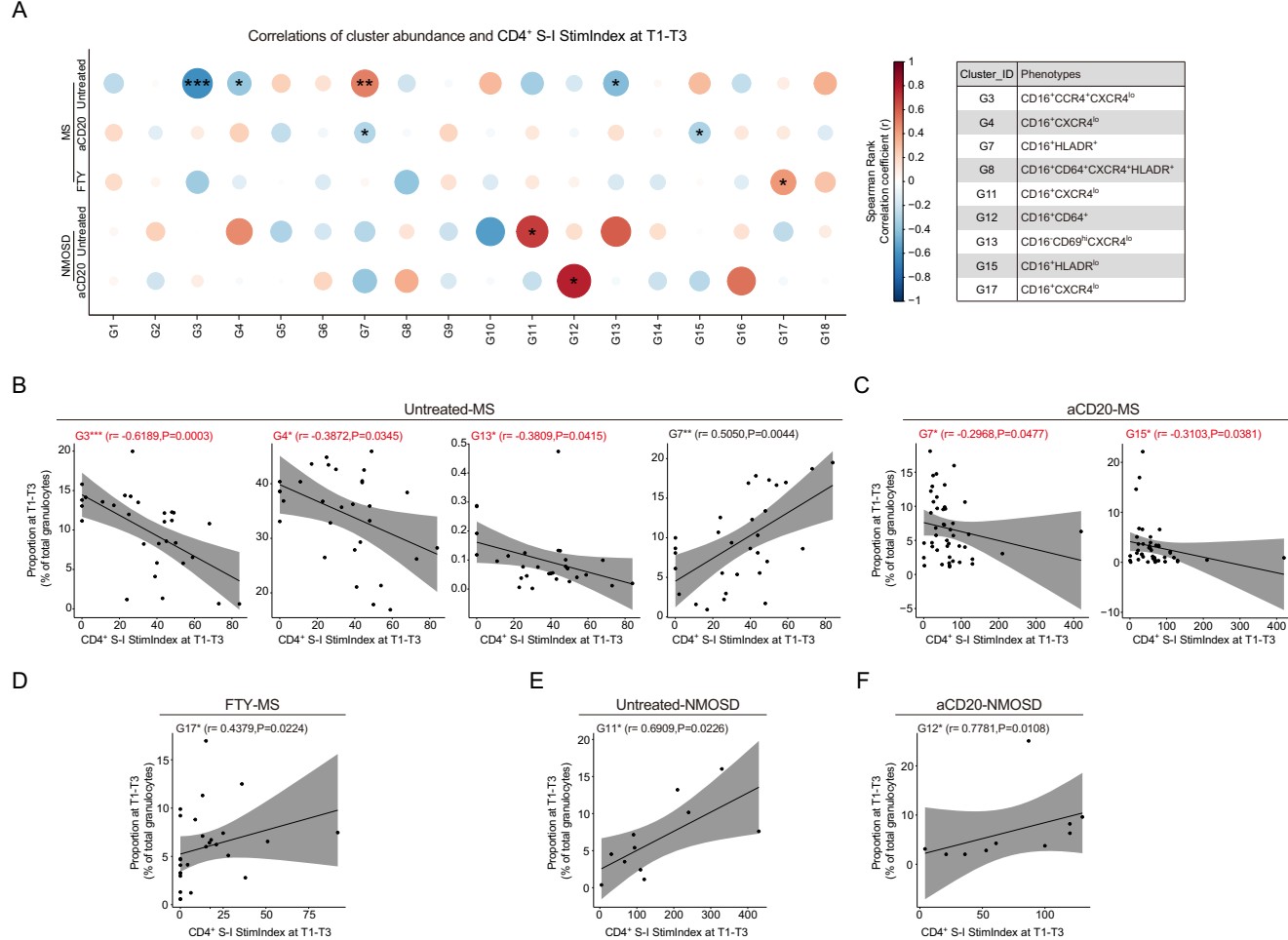

**Fig. 6 | Association of granulocyte sub-populations with S-I-specific CD4⁺ T cell responses followed primary and booster vaccination (at T1-T3) in MS and NMOSD groups. A** Heatmap of the Spearman correlation coefficients between granulocyte cluster proportion and CD4⁺ S-I Stim.Index at T1-T3 in MS and NMOSD groups. Nonparametric Spearman correlation test (r), two-sided (*p < 0.05, **p < 0.01, ***p < 0.001, ****p < 0.0001; the exact p-values are shown in **B–F**.

Correlated clusters were defined in the table. **B–F** Scatter plots showing correlation between the proportion of granulocyte cluster and CD4⁺ S-I Stim.Index at T1-T3 in Untreated-MS group **B**, aCD20-MS group **C**, FTY-MS group **D**, Untreated-NMOSD group **E**, and aCD20-NMOSD group **F**. The red text represents the negative correlation. Nonparametric Spearman correlation test (r), two-sided. Black lines and gray shadows represent the best-fitted smooth line and 95% confidence interval.

disease activity on brain/spinal MRI), (3) continuous immunomodulatory treatment or no treatment for at least 4 months, and (4) no medical indications against SARS-CoV-2 vaccinations. The exclusion criteria included: Previous SARS-CoV-2 infection, and heterologous vaccination regimes.

Medical histories, blood samples and nasopharyngeal swabs were collected at four time points, prior to (i.e., baseline; T0) and 1 month after primary vaccination (i.e., 1ˢᵗ dose; T1), up to 6 months after secondary vaccination (i.e., 2ⁿᵈ dose; T2) as well as up to 4 months after

tertiary vaccination (i.e., 3ʳᵈ dose; T3) (Fig. 1A). Whole blood specimens (including granulocytes) were analyzed by CyTOF using two antibody panels with 37 markers each (Supplementary Tables 2 and 3). Antibody panel A (*Panel A*) elucidated the spectrum of circulating immune cells and their subsets (i.e., T cells, granulocytes and other myeloid cells (i.e., monocytes and DCs), NK cells), activity-related markers and chemokine receptors, whereas antibody panel B (*Panel B*) was designed to particularly investigate all major B cell subsets in detail. In addition, we measured anti-SARS-CoV-2 spike glycoprotein 1 (S1) IgG

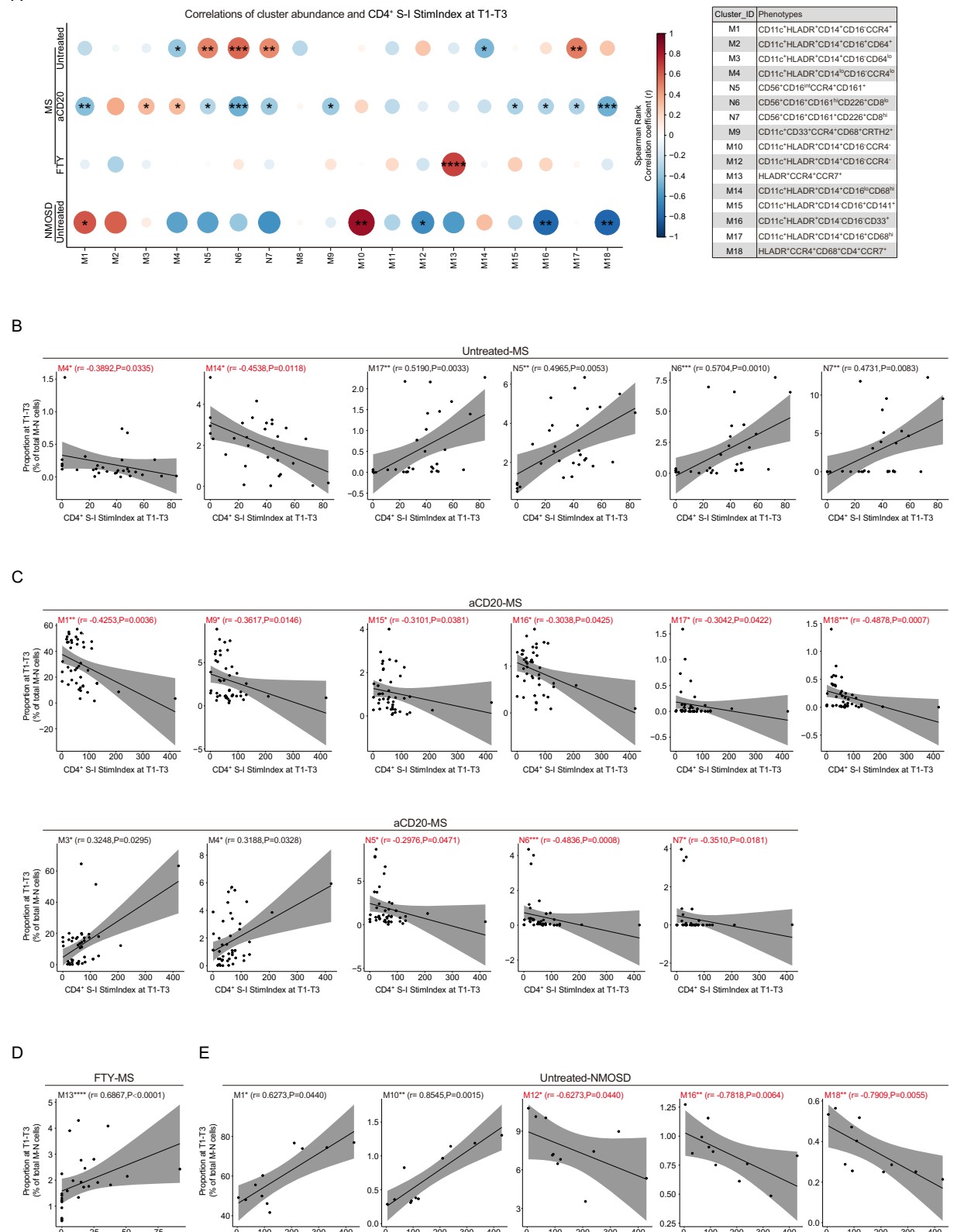

**Fig. 7 | Association of myeloid and NK cell sub-populations with S-I-specific CD4⁺ T cell responses followed primary and booster vaccination (at T1-T3) in MS and NMOSD groups.** **A** Heatmap of the Spearman correlation coefficients between MNK cell cluster proportion and CD4⁺ S-I Stim.Index at T1-T3 in MS and NMOSD groups. Nonparametric Spearman correlation test (r), two-sided (*p < 0.05, **p < 0.01, ***p < 0.001, ****p < 0.0001; the exact p-values are shown in **B**–**E**).

Correlated clusters were defined in the table. **B**–**E** Scatter plots showing correlation between the proportion of MNK cell cluster and CD4⁺ S-I Stim.Index at T1-T3 in Untreated-MS group **B**, aCD20-MS group **C**, FTY-MS group **D**, and Untreated-NMOSD group **E**. The red text represents the negative correlation. Nonparametric Spearman correlation test (r), two-sided. Black lines and gray shadows represent the best-fitted smooth line and 95% confidence interval.

and ex vivo spike-specific CD4$^+$ T cell responses Acute SARS-CoV-2 infections were ruled out by PCR testing. The number of patients of each studied group is included in every figure. Not all participants were included in the study at T0. Missing samples were due to participant unavailability or, in case of T3, due to SAR-CoV-2 infection, which was the case for one aCD20-treated MS and one aCD20-treated NMOSD patient.

### Anti-SARS-CoV-2-S1-antibody testing

Anti-SARS-CoV-2 S1 IgG was measured using a commercially available ELISA kit (Euroimmun, Cat# EI2606-9601 G) according to manufacturers' instructions[16].

### Ex vivo T cell stimulations

Freshly isolated PBMC were cultivated at 5×10$^6$ PBMC in RPMI 1640 medium (Gibco) supplemented with 10% heat inactivated AB serum (Pan Biotech), 100 U/mL penicillin (Biochrom), 0.1 mg/mL streptomycin (Biochrom). Stimulations with peptide pool covering the aa 1-643 of SARS-CoV-2 spike glycoprotein were conducted as described before[39], and were published in our previous study[16]. Briefly, PepMix SARS-CoV-2 spike glycoprotein (JPT) peptide pool (at a concentration of 1 µg/ml per peptide) in the presence of 1 µg/ml purified anti-CD28 antibody (clone CD28.2, BD Biosciences). Incubation was performed at 37 °C, 5% CO2 for 16 h with 10 µg/ml brefeldin A (Sigma-Aldrich) added after 2 h. Stimulation was stopped by incubation in 2 mM EDTA for 5 min. The "Stim.Index" defines the percentage of CD40L$^+$4-1BB$^+$ CD4$^+$ T cells among stimulated PBMCs divided by the percentage of these cells among unstimulated PBMCs.

### CyTOF analysis

**Sample processing of CyTOF-based profiling.** Whole blood (heparin) was collected from patients with MS and NMOSD at four different timepoints was fixed in SmartTube Proteomic Stabilizer as described in the user manual and stored at −80 °C until CyTOF analysis.

**Intracellular barcoding for mass cytometry.** After fixation with proteomic stabilizer, whole blood samples were thawed in Thaw/Lyse buffer and subsequently stained with premade combinations of six different palladium isotopes: $^{102}$Pd, $^{104}$Pd, $^{105}$Pd, $^{106}$Pd, $^{108}$Pd and $^{110}$Pd (Cell-ID 20-plex Pd Barcoding Kit, Fluidigm). This multiplexing kit applies a 6-choose-3 barcoding scheme that results in 20 different combinations of three Pd isotopes. After 30 min staining (at room temperature), individual samples were washed twice with cell staining buffer (0.5% bovine serum albumin in PBS, containing 2 mM EDTA). All samples were pooled together, washed, and further stained with antibodies.

**Antibodies.** Anti-human antibodies (Supplementary Table 2 for Panel A & Supplementary Table 3 for Panel B) were purchased either pre-conjugated to metal isotopes (Fluidigm) or from commercial suppliers in purified form and conjugated in house using the MaxPar X8 kit (Fluidigm) according to the manufacturer's protocol.

**Surface and intracellular staining.** After cell barcoding, washing, and pelleting, the combined samples were re-suspended in 90 µl of antibody cocktail against surface markers and incubated for 30 min at 4 °C. Then, cells were washed twice with cell staining buffer, and incubated overnight in 2% methanol-free formaldehyde solution (FA). For intracellular staining, the stained (non-stimulated) cells were subsequently washed once with staining buffer and once with permeabilization buffer (eBioscience). The samples were then stained with 100 µl of the antibody cocktails against intracellular molecules (Supplementary Tables 2 and 3) in permeabilization buffer for 30 min at room temperature. Cells were subsequently washed twice with staining buffer, then re-suspended in 1 ml iridium intercalator solution (Fluidigm) and incubated for 30 min at room temperature. Next, the samples were washed twice with cell staining buffer. Cells were pelleted and kept at 4 °C until CyTOF measurement.

**Mass cytometry data processing and analysis.** As described previously[11,40], Boolean gating was used for debarcoding. Nucleated single intact cells were manually gated according to the signals of DNA intercalators $^{191}$Ir/$^{193}$Ir and event length. For de-barcoding, Boolean gating was used to deconvolute individual samples according to the barcode combination. All de-barcoded samples were then exported as individual FCS files for further analysis. Codes used for CyTOF data analysis in this study are previously published by Crowell H et al. 2022 and available on https://github.com [https://github.com/HelenaLC/CATALYST]. Each FCS file was cleaned and compensated for signal spillover using R package *CATALYST*[41], transformed with arcsinh transformation (scale factor 5) and batch correction was implemented with a quantile normalization method to minimize batch effects[42] prior to data analysis. Prior to clustering analyses, CD19$^+$ B cells, cPARP$^-$CD66b$^+$ granulocytes, cPARP$^-$CD3$^+$ T cells and cPARP$^-$CD3$^-$CD66b$^-$CD14$^{-/+}$ MNK cells were pre-gated using FlowJo (Supplementary Figs. 1, 5 and 6). For further clustering analysis we used previously described scripts and workflows[43]. Only samples with >50 cells were considered for the downstream data analysis. For unsupervised cell population identification, we performed cell clustering with the *FlowSOM*[44] and *ConsensusClusterPlus*[45] packages using selected markers in each panel (Supplementary Tables 4 and 5). In the Panel A, we firstly identified major cell types using all markers from CD45$^+$ live cells, then took mixed populations (cluster 3, 17, and 18) out, and finally identified 18 meta-clusters on the expression of 15 selected markers, i.e., FceR1a, CCR4, CD11c, CD68, HLA-DR, CD64, CD14, CD33, CD3, CD4, CD8, CD66b, CD16, CD56, CD19. For T cell clustering, sub-clusters were first identified based on the expression of CD3, CCR7, CD4, CD8, CD226, CD69, TIGIT, CRTH2, ICOS, HLADR, CD127, CCR4, CTLA4, then six clusters with CD3$^-$ (cluster 6, 9, 12, 15, 16 and 18) were excluded, and the remaining cells were re-clustered again with 18 meta-clusters. The granulocyte compartment was clustered based on the expression of 14 markers, including FceR1a, CD45, CCR4, HLA-DR, CD64, CD33, CD95, CD66b, CD16, CD69, CXCR4, IL1beta, CD161, and CD14. We used 24 markers, including CD11c, CD47, CCR4, HLADR, CD68, CD14, CD64, CD33, CD95, CD16, CD4, CD69, TIGIT, CD141, CD206, OPN, CD127, IL-1B, CCR7, CD161, CD56, CD226, CD8, and CRTH2, to identify 18 meta clusters of MNK cells. Then took mixed population 12 out, finally confirmed 18 meta clusters. We clustered B cells excluding eight 0-cell samples with 18 meta-clusters using 26 markers in the Panel B, including CD20, IgM, CXCR3, CCR4, CD45, CD49d, HLA-DR, CD19, CD1c, CD38, IgA, CD138, IgD, IgGK, IgGL, CD62L, CD24, CXCR4, CHI3L1, CD25, CD123, CD45RO, CD27, Ki67, CD11c, Tbet. We next removed cluster 16 and 17 (CD11c$^{hi}$HLADR$^{hi}$CD123 + ) out and re-clustered again with 18 meta-clusters. The number of metaclusters used for further analysis was identified based on the delta area plots (which asses the "natural" number of clusters that best fits the complexity of the data) together with visual inspection on the phenotypic heatmap with the aim to select a cluster number with consistent phenotypes that would also allow to explore small populations. For dimensionality-reduction visualization we generated UMAP representations using all markers as input and down-sampled to a maximum of 1000 cells per sample.

**Data analysis.** Statistical analysis was performed using GraphPad Prism (version 8.0.2). Continuous variables were expressed as mean and standard deviation with or without range. Owing to the limited sample sizes, Kruskal-Wallis test followed by Dunn's correction for multiple comparisons test was used to analyze cluster abundance across different MS groups. Based on IgG status, patients were divided

into two groups: "IgG-" and "IgG + ". "IgG-" was defined as an anti-S1 IgG level below the cut-off of 1.1 optical density ratio for a positive result, while "IgG + " was defined as anti-S1 IgG level above this cut-off. Significant differences in cluster abundance between IgG status were calculated using Mann–Whitney's U-test. Spearman's correlation coefficients were used to evaluate the correlation between cluster abundance and anti-S1 IgG level/CD4 + S-I StimIndex. $P < 0.05$ was considered statistically significant.

## Reporting summary
Further information on research design is available in the Nature Portfolio Reporting Summary linked to this article.

## Data availability
The CyTOF data generated in this study have been deposited in the FlowRepository database under accession code FR-FCM-Z7ZZ [http://flowrepository.org/id/FR-FCM-Z7ZZ]. Source data are provided with this paper as Source Data file. Source data are provided with this paper.

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

## Acknowledgements

We thank Nicolina Schmidt-Sandte for excellent technical assistance with sample collection. We would also like to acknowledge the assistance of the BIH Cytometry Core Facility (Charité—Universitätsmedizin Berlin, Germany). M.W. received the PhD scholarship from the Chinese Scholarship Council (CSC). C.B. was funded by the Deutsche Forschungsgemeinschaft (DFG, the German Research Foundation—Project-ID 259373024—CRC/TRR 167 (B05)). J.B., L.L., and A.T. were funded by the Federal Ministry of Health through a resolution of the German Bundestag (Charité Corona Cross (CCC) 2.0 and 2.1 and Charité Corona Protect (CCP)). J.B. was partially funded by the DFG as part of the clinical research unit (CRU339: Food allergy and tolerance (FOOD@)—409525714.

## Author contributions

C.B., L.M.-A. and F.P. conceived and designed the project. C.B., C.F.Z. and D.K. designed the antibody panels for mass cytometry. T.S.-H., J.B.-S., M.G. and A.K. recruited the patients and provided the patients´ clinical data. L.E.S., J.B., L.L. and A.T. performed the analyses of anti-S1 IgG level and S-I-specific CD4+ T cell reactivity. M.W., A.D., G.G., C.F.Z. and D.K. performed CyTOF experiments and data analyses. C.B., L.M.-A., G.G., M.W., S.S. and B.S. analyzed and interpreted the data. C.B., L.M.-A., F.P., M.W., A.D., C.F.Z., S.S., and B.S. wrote the manuscript.

## Funding

## Competing interests

F.P. received research support for this study from F. Hoffmann-La Roche Ltd., Alexion Pharma Germany GmbH, and Horizon Therapeutics Ireland DAC. The remaining authors declare no competing interests.

## Additional information

[1]Experimental and Clinical Research Center, a cooperation between the Max Delbrück Center for Molecular Medicine in the Helmholtz Association and Charité—Universitätsmedizin Berlin, Berlin, Germany. [2]Charité—Universitätsmedizin Berlin, corporate member of Freie Universität Berlin and Humboldt-Universität zu Berlin, Berlin, Germany. [3]Max Delbrück Center for Molecular Medicine in the Helmholtz Association (MDC), Berlin, Germany. [4]Neuroimmunology and Multiple Sclerosis Unit and Laboratory, Sourasky Medical Center, Tel Aviv Sourasky Medical Center, Tel Aviv, Israel. [5]Department of Infectious Diseases, Respiratory Medicine and Critical Care, Charité—Universitätsmedizin Berlin, corporate member of Freie Universität Berlin, Humboldt-Universität zu Berlin, and Berlin Institute of Health, Berlin, Germany. [6]Institute of Medical Immunology, BIH Center for Regenerative Therapies, Charité—Universitätsmedizin Berlin, and Berlin Institute of Health Berlin, Berlin, Germany. [7]Flow&MassCytometry Core Facility, Berlin Institute of Health at Charité—Universitätsmedizin Berlin, Berlin, Germany. [8]Neuroscience Clinical Research Center, Charité—Universitätsmedizin Berlin, corporate member of Freie Universität Berlin, Humboldt-Universität zu Berlin, and Berlin Institute of Health, Berlin, Germany. [9]Translational Immunology, Berlin Institute of Health at Charité—Universitätsmedizin Berlin, Berlin, Germany. [10]Faculty of Medicine and Sagol School of Neuroscience Tel Aviv University, Tel Aviv, Israel. [11]Si-M / "Der Simulierte Mensch" a science framework of Technische Universität Berlin and Charité—Universitätsmedizin Berlin, Berlin, Germany. [12]Berlin Institute of Health (BIH) at Charité—Universitätsmedizin Berlin, Immunomics—Regenerative Immunology and Aging, Berlin, Germany. [13]Department of Neurology with Experimental Neurology, Charité—Universitätsmedizin Berlin, corporate member of Freie Universität Berlin, Humboldt-Universität zu Berlin, and Berlin Institute of Health, Berlin, Germany. [14]These authors jointly supervised this work: Friedemann Paul, Lil Meyer-Arndt, Chotima Böttcher. ✉e-mail: chotima.boettcher@charite.de

