## [Peer Review File · Nature Communications]

REVIEWER COMMENTS

Reviewer #1 (expert in neurology, MS, NMOSD):

This is an interesting study which can improve our knowledge regarding of the effect of DMDs on our immune system's responses to different antigens. But I believe the authors should answer some questions and revise their manuscript according to them before publication:

- 1- NMOSD is an astrocytopathy and it differs from MS and MOGAD pathologically. It should be revised in the manuscript.
- 2- The focus of this study was on SARS-COVID- vaccine. What can this study tell us about other infections and vaccines?
- 3- The SARS-COVID 19 is not serious as it was previously. The authors should add a separate section discussing how their study can be important for other infections and vaccines.
- 4- We know that some DMDs including fingolimod and anti-CD20 can increase the risk of infections in patients with MS. Can these DMDs decrease the rate of special infections according to this study?
- 5- What is the practical message of this study? Should the physicians check myeloid cell compartment routinely?
- 6- The small sample size of patients with MOGAD can affect the authors' significantly.
- 7- What were the exclusion and inclusion criteria?
- 8- The possible comorbidities of these patients which can affect the results have not been stated in the manuscript.
- 9- The history of recent infection or vaccination is important.

Reviewer #2 (expert in mass cytometry, single-cell analysis, SARS-CoV2):

Overall comment:

Wang et al. used mass cytometry to investigate changes in circulating immune cells in multiple sclerosis patients after treatment with various disease-modifying therapies and COVID-19 vaccinations. A lot of interesting data are presented, but they were difficult to follow. It was unclear what the major finding(s) was.

It is an interesting study but needs some polishing to better clarify the story. The methodology and analyses were sound with only minor comments. Care should be taken when interpreting the results as

commented. Although it requires major revision, it has the potential to be a great paper that would be ideal for the Nat Comm audience.

Major comments:

- Introduction is too long and should be more succinct. There are a lot of details provided, but it's not clear how they all relate to each other. The second and third paragraphs can be split as they're very long.
- The colour scheme for each figure needs changing. Don't use different tones of the same colour, it's very hard to differentiate between groups. This is most evident in Fig2 and the PCA plots, as it's not clear which group each circle belongs to.
- It wasn't clear why Fig2 only looked at anti-CD20 and no other DMT. Fig1 already showed how many clusters differed after anti-CD20 treatment (at baseline), so what does Fig2 provide? When looking at other timepoints, the anti-CD20 treatment will have long-lasting effects so are the differences between timepoints the vaccination or anti-CD20? It would make more sense comparing between timepoints for each treatment. I'd consider removing Fig2, otherwise provide more justification for these comparisons. It's also possible to do some statistical tests to compare between the groups on the PCA, including ANOVA or PERMANOVA to further support differences.
- There's a large range of treatment duration before vaccination across patients (SuppTable1). Is it fair to compare a patient that has been on treatment for 2 months to someone after 233 months? Provide more details on the range for each DMT. Majority of DMT will have long-lasting effects, but the largest changes occur shortly after initial treatment.
- The primary readout are proportions, so when one cluster decreases another must decrease. This is a major limitation of the study, as absolute numbers would be ideal. I understand this may not be possible, but it is important to discuss as the changes observed may be a result of a difference in one cluster artificially altering the results of another cluster.
- Discussion paragraphs are far too long so the major points are lost. The first paragraph repeats what the introduction stated (namely the effect of DMT on innate cells), so can be cut down.
- It is not clear what the major take-home message is. Comparisons were made between the different treatments and then there were correlations with COVID-related antibodies and specific T-cell subsets. Although there were several timepoints collected, most comparisons only used T0 or combined T1-3. Why not look at cluster levels over time for each group? I appreciate the patient numbers will be small, but it would provide some insight into DMT-, vaccine-, and disease-related changes.

Minor comments:

- In second paragraph of results, state the clustering algorithm used that identified 18 clusters
- Be careful with the term "frequency", as you're showing "percentages" which may be a better term to use (or proportions)

- In Fig1F, make the order of populations the same as Fig1D for consistency
- On page 7, the components of the PCA are “driving the differences between patients”. The components are based on overall variability and not just differences between groups. This paragraph should be rephrased to reflect this.
- SuppFig5 and SuppFig6 add colour legend for each treatment
- Page 8, “Therefore, the presence of these B cell subsets was crucial for robust vaccine- specific IgG antibody production” is too strong a statement. The levels of these clusters were associated with IgG levels and not necessarily responsible for them.
- Page 9 “G5 and G11 strongly contributed to treatment-mediated differences across all post-vaccination time points” is incorrect. These clusters had high variance which is why they were highlighted by PCA, and happen to be different between groups. This needs rephrasing.
- In Fig3F, what do the “IgG+” and “IgG–” groups represent? Is there a cut-off for anti-S1 IgG levels to differentiate between them? Make sure this is stated in the results for clarification.
- SuppFig8, what do the colours represent? And what are the boxplots showing (median/mean/etc.)?
- Fig5H-I, be consistent (with or without confidence intervals) with other correlation plots in other figures.
- Fig6, what does the red text indicate?
- Page 14, “were positively... or inhibit” should be “were positively... or negatively”

Reviewer #3 (expert in neurology, MS, NMOSD):

Thank-you for asking me to review this interesting manuscript

-What are the noteworthy results?

This demonstrates that predominantly DMTs have an impact on the immune system and describes such changes in NMO and MS and small group of MOG subjects. The authors identify the impairment with B cell therapies but a wider impact on the immune system which is novel but not unexpected and needs to be characterised.

Will the work be of significance to the field and related fields?

The work describes changes with vaccination but does not determine the effects of vaccination as it is not unexpected that DMTs impact B cells. To influence the field we need to understand how such changes effect infection rates/susceptibility/severity of disease.

How does it compare to the established literature? If the work is not original, please provide relevant references.

It does extend the literature in describing the change in other parts of the immune systems beyond B cells.

The author comment on prior work where DMTs impact the immune system and where they have extended the work in offering detailed immune signatures in the innate and adaptive arm.

Does the work support the conclusions and claims, or is additional evidence needed?

The conclusion to recommend innate monitoring based on a study of this size is not credible. We do not know how the vaccination changes or not impact serious infection rates. This is required to institute clinical monitoring. In many cases in current practice B cell monitoring is not carried out.

Are there any flaws in the data analysis, interpretation and conclusions? Do these prohibit publication or require revision?

There is a very small group of MOGAD subjects limiting any conclusion about them. One might consider removing them.

In addition the longitudinal analysis has a number of missing subjects that is unexplained especially if they are at risk of infections eg what happened to the dropouts? did anyone get COVID?

The missing subjects need to be described and added as a limitation.

Is the methodology sound? Does the work meet the expected standards in your field?

The laboratory work is well performed and appears methodologically sound.

There is no pre-specified protocol mentioned. The numbers are limited. The study in its current form is not powered nor designed appropriately to help inform us of the impact of the cellular changes identified.

Is there enough detail provided in the methods for the work to be reproduced?

I cannot comment on all the techniques as I have not used them all.

REVIEWER COMMENTS

Reviewer #1 (expert in neurology, MS, NMOSD):

This is an interesting study which can improve our knowledge regarding of the effect of DMDs on our immune system's responses to different antigens. But I believe the authors should answer some questions and revise their manuscript according to them before publication:

Response:

We thank the reviewer for the positive feedback and valuable suggestions to improve our manuscript. We have revised the manuscript accordingly.

1- NMOSD is an astrocytopathy and it differs from MS and MOGAD pathologically. It should be revised in the manuscript.

Response:

We have revised the introduction to emphasize the differences between NMOSD and MS (page 3). Also, in response to the reviewer's comment #6 and to the comment of the reviewer #3, we have made the decision to exclude the MOGAD patients from our analyses. By doing so, we aimed to prevent any potential statistical biases that could arise from the small sample size. As a result, we have removed any mention of MOGAD from our manuscript to ensure clarity and avoid confusion.

2- The focus of this study was on SARS-COVID- vaccine. What can this study tell us about other infections and vaccines?

Response:

See below.

3- The SARS-COVID 19 is not serious as it was previously. The authors should add a separate section discussing how their study can be important for other infections and vaccines.

Response (#2 and #3):

Following the reviewer's suggestion, we have incorporated potential implications of our findings for other infections and vaccines into the discussion section (page 12, 15 and 16). Regardless of the specific infection or vaccine, antigens/pathogens stimulate cellular and humoral immune responses through similar mechanisms. Thus, studying immune responses to (only) one antigen/pathogen can provide valuable insights into broader immune responses to other antigens/pathogens (such as infections and vaccines). In our study, we specifically evaluated cellular and humoral responses to primary and booster vaccinations under different DMTs for MS and NMOSD.

We identified granulocyte populations that exhibited alterations in their proportion after treatment with DMTs and further characterized these populations in relation to cellular and humoral responses following vaccination. Our findings shed light on the heterogeneity and the role of granulocytes, which are often considered homogenous and underestimated, in immune responses to vaccination. This involvement of granulocytes and other myeloid cells becomes

more prominent when the T and B cell compartments are compromised, such as under treatment with DMTs. Of note, neutropenia, a rare complication observed in patients treated with anti-CD20 antibody therapies including those with MS, can increase the risk of infection (Rauniyar et al. 2022 (PMID: 35079395); Cohen et al. 2019 (PMID: 30635476); Malpica Castillo et al. 2020 (PMID: 32017201)). Thus, in addition to their roles in infections, granulocytes may also have a role for other vaccines. Monitoring granulocytes alongside lymphocyte populations may aid physicians in better assessing a patient's immune status. Furthermore, in this study, we demonstrated a simple sample preparation protocol for whole blood samples (i.e., without PBMCs isolation) and CyTOF analysis to characterize granulocytes and other PBMCs, which may encourage more neuroimmunologist to study this cell population in the context of infections and vaccines.

4- We know that some DMTs including fingolimod and anti-CD20 can increase the risk of infections in patients with MS. Can these DMTs decrease the rate of special infections according to this study?

Response:

DMTs such as anti-CD20 antibodies and fingolimod have been observed to decrease the number of B cells (anti-CD20) or B and T cells (fingolimod) in the peripheral blood and can also affect the composition of myeloid cells. Altogether, this increases the susceptibility to infections. However, our findings suggest that under anti-CD20 antibody therapy, myeloid cells may assume a crucial role in pathogen clearance using their phagocytosis and cytokine-producing capabilities, antigen-presenting cell function and anti-inflammatory properties (see page 12-13 in the revised manuscript).

5- What is the practical message of this study? Should the physicians check myeloid cell compartment routinely?

Response:

We thank the reviewer for raising this important question, which has prompted us to further emphasize the clinical implications of our study (see the Discussion section, page 12-13).

Based on our findings, we can summarize the main messages as follows:

1. DMTs such as anti-CD20 and fingolimod not only impact T and B lymphocytes but also have significant effects on the composition of granulocytes and other myeloid cells. These effects vary depending on the specific DMT and may also differ between diseases.
2. Granulocytes, previously considered a homogenous cell population, can be subdivided into distinct sub-populations based on marker expression profiles. These sub-populations show differential associations with cellular and humoral responses to vaccines.

While we cannot directly prove the specific functions of individual granulocyte populations in infections and vaccines due to technical limitations, our findings strongly suggest the importance of monitoring the myeloid cell compartment, including granulocytes, in patients receiving DMTs. This is particularly crucial in the context of infections or vaccines. As previously mentioned, DMT-treated patients may experience neutropenia, which significantly compromises the host's defense system. Therefore, alongside monitoring T and B lymphocytes, it is necessary to also assess innate immune cells to gain a comprehensive understanding of the patient's immune status. Considering the potential risk associated with infections and the impact of DMTs on myeloid cells, routine monitoring of the myeloid compartment may be warranted in certain clinical situations. However, the decision should be

based on individual factors such as the specific DMT, the patient's disease profile, and their risk factors.

6- The small sample size of patients with MOGAD can affect the authors' significantly.

Response:

In response to the reviewers' suggestions, as well as the comment of the reviewer #3, we have removed the samples from the MOGAD patients from our study. Consequently, all data had to be re-analyzed, including clustering and statistical testing, without the MOGAD patients. This re-analysis resulted in different UMAP plots and cluster IDs compared to the original manuscript. However, the overall findings remained unchanged. As a result, all figures were newly generated reflecting the outcomes of the new data analysis.

7- What were the exclusion and inclusion criteria?

Response:

We have described the exclusion and inclusion criteria in the methods section of the revised manuscript (page 16 and 17).

Inclusion criteria were: (1) MS diagnosis according to the McDonald criteria of 2017 or NMOSD diagnosis according to Wingerchuk (2015), (2) stable disease for at least 3 months (no acute relapse therapy, no clinical progression or new symptoms suggestive of relapse, no disease activity on brain/spinal MRI), (3) continuous immunomodulatory treatment or no treatment for at least 3 months, and (4) no medical contraindications against SARS-CoV-2 vaccination.

The exclusion criteria included: Previous SARS-CoV-2 infection, and heterologous vaccination regimes.

8- The possible comorbidities of these patients which can affect the results have not been stated in the manuscript.

Response:

We agree with the reviewer that possible comorbidities are another critical issue. Although none of the participants in our study had comorbidities of the immune system, it is important to consider that other diseases, such as cardiovascular or metabolic diseases, could potentially affect immune responses. We have addressed this limitation in the discussion section of the revised manuscript (page 15-16).

9- The history of recent infection or vaccination is important.

Response:

We agree with the reviewer. Previous SARS-CoV-2 infections were an exclusion criterion of this study (page 17). We provide a timeline, which illustrates the sampling time points in timely relation to the participants' COVID vaccinations, in Fig. 1A.

Reviewer #2 (expert in mass cytometry, single-cell analysis, SARS-CoV2):

Wang et al. used mass cytometry to investigate changes in circulating immune cells in multiple sclerosis patients after treatment with various disease-modifying therapies and COVID-19 vaccinations. A lot of interesting data are presented, but they were difficult to follow. It was unclear what the major finding(s) was. It is an interesting study but needs some polishing to better clarify the story. The methodology and analyses were sound with only minor comments. Care should be taken when interpreting the results as commented. Although it requires major revision, it has the potential to be a great paper that would be ideal for the Nat Comm audience.

Response:

We thank the reviewer for the insightful feedback and valuable suggestions to improve our manuscript. We have revised the manuscript accordingly.

Major comments:

• Introduction is too long and should be more succinct. There are a lot of details provided, but it's not clear how they all relate to each other. The second and third paragraphs can be split as they're very long.

Response:

We appreciate the reviewer's feedback regarding the length and organization of the introduction. We have revised the manuscript to ensure that the paragraphs are appropriately divided for better readability and coherence and the introduction is easier to follow and more succinct.

• The colour scheme for each figure needs changing. Don't use different tones of the same colour, it's very hard to differentiate between groups. This is most evident in Fig2 and the PCA plots, as it's not clear which group each circle belongs to.

Response:

We thank the reviewer for this suggestion and have changed the color palette of our figures.

• It wasn't clear why Fig2 only looked at anti-CD20 and no other DMT. Fig1 already showed how many clusters differed after anti-CD20 treatment (at baseline), so what does Fig2 provide?

Response:

We thank the reviewer for this question and have now revised the text to improve clarity. Briefly, in Fig. 1, we presented the differentially abundant clusters observed in the baseline samples of different treatment groups. Notably, significant changes were observed in aCD20- and FTY-treated patients.

In Fig. 2, we aimed to highlight the overall differences between untreated and aCD20-treated patients in both MS and NMOSD patient cohorts at different time points following vaccination. Our findings revealed that, even after vaccination, a similar set of clusters (compared to baseline) exhibited high variance among all groups. As there are no FTY-treated NMOSD patients, we focused our analysis on the aCD20-treated groups in Fig. 2. In addition, Supplementary Fig. 2 presents the PCA plots of all treated and untreated groups, which again showed that the similar set of clusters drove the observed differences between each sample.

Together with the results shown in Suppl. Fig. 3 and 4, our findings suggest that overall variability between samples was primarily driven by treatment rather than vaccination.

When looking at other timepoints, the anti-CD20 treatment will have long-lasting effects so are the differences between timepoints the vaccination or anti-CD20? It would make more sense comparing between timepoints for each treatment. I'd consider removing Fig2, otherwise provide more justification for these comparisons. It's also possible to do some statistical tests to compare between the groups on the PCA, including ANOVA or PERMANOVA to further support differences.

Response:

We appreciate the reviewer's valuable suggestion. We understand the importance of Fig. 2 in providing an overview of the defined clusters with high variance that drive the observed differences between samples over time. Therefore, we have decided to retain Fig. 2 as presented in the original manuscript. However, as suggested, we have revised the text to provide additional justification for the comparisons and their relevance. Additionally, we have included new plots in Supplementary Fig. 3, depicting cell proportions of each treatment group at different time points, and Supplementary Fig. 4, illustrating the cell proportions at each time point with different treatments. These new plots contribute to the clarity of the findings, demonstrating that the changes in immune profiles are mainly the result of the long-lasting effects of DMTs. Furthermore, we observed no significant differences between time points within each DMT-treated group and consequently combined the data from the T1-T3 time points for a comprehensive analysis.

• There's a large range of treatment duration before vaccination across patients (SuppTable1). Is it fair to compare a patient that has been on treatment for 2 months to someone after 233 months? Provide more details on the range for each DMT. Majority of DMT will have long-lasting effects, but the largest changes occur shortly after initial treatment.

Response:

We thank the reviewer for this important comment. In Suppl. Table 1 of the original manuscript, we did not differentiate between treatments but provided overall information for all MS and NMOSD patients together. To provide readers with a better understanding of our results, we have now included detailed information about the treatment duration for each patient and each specific treatment in Suppl. Table 6.

As previously reported in our own study (Meyer-Arndt et al., 2022; <http://dx.doi.org/10.1136/jnnp-2022-329395>), it is known that FTY treatment duration in MS patients negatively correlates with (neutralizing) antibody induction. In our current study, we observed a similar trend, where treatment duration showed a significantly negative correlation with the proportion of B and NK cells, as well as certain immune cell sub-populations, including granulocytes, as illustrated in the figure below.

Figure 1 Correlation between proportion at T0 (%) of **A)** NK (C10 & C16) and B cells (C15) as well as **B)** subpopulations of granulocytes, **C)** B cells, **D)** T cells and **E)** MNK cells.

The aim of our current study is to evaluate the correlation between anti-S1 IgG levels and the proportion of immune cell sub-populations, with a particular emphasis on myeloid cells. These two parameters are known to be influenced by treatment duration. Therefore, we believe it is valid to include a wide range of treatment durations in our correlation analyses. By doing so, we aim to gain a comprehensive understanding of the associations between these factors and provide valuable insights into the impact of treatment duration on immune responses.

• *The primary readout are proportions, so when one cluster decreases another must decrease. This is a major limitation of the study, as absolute numbers would be ideal. I understand this may not be possible, but it is important to discuss as the changes observed may be a result of a difference in one cluster artificially altering the results of another cluster.*

Response:

We absolutely agree with the reviewer's observation that interpreting the results based solely on changes in proportion can be misleading. It is important to consider that an "increased" cluster may not necessarily indicate an actual increase, but rather a relative prominence due to depletion in other populations. This becomes particularly evident when comparing different cell populations, as shown in Figure 1, where B and T cells are depleted, leading to the apparent dominance of myeloid cells. However, in this current study, we specifically compare the cellular composition within a single population such as granulocytes (as shown in Figure 3). In these cases, the proportion of sub-populations is calculated based on the total number of cells within that specific population (e.g., the proportion (%) of G1 within total granulocytes). Consequently, the proportion of sub-populations within each cell type is not affected by other cell types. To address this point and ensure clarity, we have revised the text (page 7) accordingly, providing a clear explanation of the approach and its implications.

• *Discussion paragraphs are far too long so the major points are lost. The first paragraph repeats what the introduction stated (namely the effect of DMT on innate cells), so can be cut down.*

Response:

We thank the reviewer for this important point and have revised the discussion accordingly.

• *It is not clear what the major take-home message is. Comparisons were made between the different treatments and then there were correlations with COVID-related antibodies and specific T-cell subsets. Although there were several timepoints collected, most comparisons only used T0 or combined T1-3. Why not look at cluster levels over time for each group? I appreciate the patient numbers will be small, but it would provide some insight into DMT-, vaccine-, and disease-related changes.*

Response:

We thank the reviewer for this comment. In response, we have revised the manuscript to emphasize our findings, as suggested. In summary, Briefly, our study highlights that changes in immune cell compositions vary across different diseases and treatments. Specifically, certain innate immune cell clusters, which are altered after DMTs, such as CD64⁺ or CXCR4⁺ or HLADR⁺ granulocytes, show significant correlations with immune responses to vaccination, suggesting their involvement in this process. While our study does not establish their precise functions, it is intriguing to hypothesize that these defined clusters exhibit distinct functions depending on the specific disease and treatment context. Our findings offer novel insights into the myeloid cell compartment and its correlation with vaccination responses. Furthermore, considering the potential development of neutropenia in a subset of certain DMT-treated patients, monitoring the myeloid cell compartment alongside lymphocytes may prove valuable in clinical decision-making. As mentioned earlier, we have now included new plots depicting the cell proportion for each treatment at different time points (Suppl. Fig. 3) and at each specific time point with different treatments (Suppl. Fig. 4). In the original manuscript, we combined the data from the T1-T3 time points, as there were only few significant differences observed between time points within each DMT-treated group, as shown below:

Figure 2 Changes in proportion of cell population overtime: **A)** CD8⁺ T cells in MS patients treated with FNβ and classical monocytes in aCD20-treated NMOSD patients. **B)** granulocyte subpopulations in MS patients treated with FNβ and **C)** T cell sub-population in aCD20-treated MS patients.

Minor comments:

- *In second paragraph of results, state the clustering algorithm used that identified 18 clusters*

Response:

We have added the missing information to the main text (page 5).

- *Be careful with the term “frequency”, as you’re showing “percentages” which may be a better term to use (or proportions)*

Response:

We have changed “frequency” to “proportion” as suggested.

- *In Fig1F, make the order of populations the same as Fig1D for consistency*

Response:

The figure has been revised as suggested.

- *On page 7, the components of the PCA are “driving the differences between patients”. The components are based on overall variability and not just differences between groups. This paragraph should be rephrased to reflect this.*

Response:

We thank the reviewer for this correction. We have revised the paragraph (page 6) accordingly.

- *SuppFig5 and SuppFig6 add colour legend for each treatment*

Response:

The figures have been revised as suggested.

- *Page 8, “Therefore, the presence of these B cell subsets was crucial for robust vaccine-specific IgG antibody production” is too strong a statement. The levels of these clusters were associated with IgG levels and not necessarily responsible for them.*

Response:

We thank the reviewer for bringing this to our attention. As suggested, we have revised the text (page 7) to address this concern and avoid over-estimation.

- *Page 9 “G5 and G11 strongly contributed to treatment-mediated differences across all post-vaccination time points” is incorrect. These clusters had high variance which is why they were highlighted by PCA, and happen to be different between groups. This needs rephrasing.*

Response:

We thank the reviewer for this important correction. We have rephrased the relevant sentence (page 8) as suggested.

• *In Fig3F, what do the “IgG+” and “IgG–“ groups represent? Is there a cut-off for anti-S1 IgG levels to differentiate between them? Make sure this is stated in the results for clarification.*

Response:

We apologize for any confusion caused by the unclear definition. To clarify, the “IgG-” group refers to patients with an anti-S1 IgG level below the cut-off for a positive result, indicating the absence of detectable anti-S1 IgG antibodies. On the other hand, the “IgG+” group represents patients with an anti-S1 IgG level above the cut-off. We have now included this information in the figure legend of the revised figure.

• *SuppFig8, what do the colours represent? And what are the boxplots showing (median/mean/etc.)?*

Response:

In Supplementary Fig. 8 (now Supplementary Fig. 10 in the revised manuscript), the red bars represent the clusters whose proportion shows positive correlation with anti-S1 IgG levels. Conversely, the light blue bars represent clusters that exhibit negative correlations with anti-S1 IgG levels. These plots display the median expression of the defined markers for each granulocyte cluster. We have revised the figure and its corresponding legend by incorporating the missing information to enhance clarity.

• *Fig5H-I, be consistent (with or without confidence intervals) with other correlation plots in other figures.*

Response:

We have checked and ensured the consistency of all figures.

• *Fig6, what does the red text indicate?*

Response:

We apologize for the previous lack of clarity in the illustration. In order to improve readability, we have made changes to clearly distinguish between positive and negative correlations. We now use different colors to indicate the correlations, with red text representing negative correlations and black text representing positive correlations. This information has been added to the figure legends.

• *Page 14, “were positively... or inhibit” should be “were positively... or negatively”*

Response:

We have revised this sentence accordingly.

Reviewer #3 (expert in neurology, MS, NMOSD):

Thank-you for asking me to review this interesting manuscript

-What are the noteworthy results?

This demonstrates that predominantly DMTs have an impact on the immune system and describes such changes in NMO and MS and small group of MOG subjects. The authors identify the impairment with B cell therapies but a wider impact on the immune system which is novel but not unexpected and needs to be characterized.

Response:

We thank the reviewer for this helpful feedback and the recognition of our study results, which emphasize the effects of DMTs and primary as well subsequent antigen encounter on immune cell composition in patients with NMOSD and MS. While changes in immune cell composition are not surprising, our study provides valuable insights into the diverse nature of these changes across different treatments and diseases. Specifically, we uncover intriguing alterations within the innate immune cell compartment, including CD64⁺ granulocytes, CXCR4⁺ granulocytes, and CD8^{hi} NK cells, alongside the lymphocyte compartments. These findings offer novel perspectives on the complex interplay between DMTs and immune cell composition. Moreover, our study reveals potential correlations between certain immune cell changes and vaccine responses, highlighting their relevance in the context of vaccination or first-time infection and adding depth to our understanding of the immunological landscape in DMT-treated patients with neuroimmunological diseases.

Will the work be of significance to the field and related fields?

The work describes changes with vaccination but does not determine the effects of vaccination as it is not unexpected that DMTs impact B cells. To influence the field we need to understand how such changes affect infection rates/susceptibility/severity of disease.

Response:

As described earlier, our study focuses on characterizing changes in the innate immune cell compartment in patients treated with DMTs before and after vaccination, alongside the investigation of B and T lymphocytes. This approach sheds new light on the importance of monitoring myeloid cells in these patients, which is a novel consideration. We therefore believe that our findings will have a significant impact on the field of neuroimmunology and related disciplines.

However, we agree with the reviewer that further investigation is necessary to understand how these changes in immune cell composition can affect infection rates, susceptibility, and severity of disease. To address this, additional approaches are needed, including *in vitro* functional studies and systematic longitudinal studies to observe potential long-term effects. Moreover, different patient cohorts will be required to study the influence of these changes on infection rates, considering that our current study excluded patients with previous or active infections. We have now addressed this aspect in our discussion section.

Nonetheless, our findings provide a valuable platform for the characterization of multiple immune cell populations, particularly granulocytes, in the context of neuroimmunological diseases, serving as a foundation for future research.

How does it compare to established literature? If the work is not original, please provide relevant references.

It does extend the literature in describing the change in other parts of the immune systems beyond B cells. The author comments on prior work where DMTs impact the immune system and where they have extended the work in offering detailed immune signatures in the innate and adaptive arm.

Response:

We appreciate the reviewer's recognition of the novelty and significance of our study, which aims to extend our knowledge of the diverse effects of DMTs on the human immune system.

Does the work support the conclusions and claims, or is additional evidence needed? The conclusion to recommend innate monitoring based on a study of this size is not credible. We do not know how the vaccination changes or not impact serious infection rates. This is required to institute clinical monitoring. In many cases in current practice B cell monitoring is not carried out.

Response:

We appreciate the reviewer's important feedback, and we agree that the findings from our study, based on a relatively small patient cohort, are not sufficient to draw conclusive recommendations regarding the routine clinical practice of monitoring innate cells. To obtain robust and definitive conclusions in the context of human patients, a study involving thousands of patients would be necessary. However, conducting such a large-scale study presents challenges, primarily due to the cost and complexity of data analysis. Therefore, a study of a larger patient cohort would typically require a shift towards lower dimensional analysis (e.g., instead of examining > 60 markers at the single-cell level, a reduced panel of 5 to 8 markers). We have included this highly relevant aspect in our discussion section (page 16).

We acknowledge the limitations in terms of sample size and scope of our study, and it is important to take them into consideration when interpreting our results. However, despite these constraints, our study successfully identified a set of markers and sub-populations that are involved in immune responses to DMTs and vaccinations. These well-defined parameters provide a valuable foundation for future investigations involving much larger cohorts, with a focus on lower-dimensional immune cell monitoring approaches.

Are there any flaws in the data analysis, interpretation and conclusions? Do these prohibit publication or require revision?

There is a very small group of MOGAD subjects limiting any conclusion about them. One might consider removing them.

Response:

We agree with the reviewer's observation regarding the small sample size of the MOGAD cohort, which aligns with the feedback from reviewer #1 and #3. We have thus made the decision to exclude the MOGAD patients from our analyses to avoid potential statistical biases that could arise from the small sample size. Consequently, we had to conduct a reanalysis of all the data, including clustering and statistical testing, without the MOGAD patients. This reanalysis resulted in different UMAP plots and cluster-IDs compared to the original

manuscript. However, it is important to note that the overall findings remained consistent and unchanged. Furthermore, all figures were newly generated reflecting the outcomes of the new data analysis.

In addition the longitudinal analysis has a number of missing subjects that is unexplained especially if they are at risk of infections eg what happened to the dropouts? did anyone get COVID?

The missing subjects need to be described and added as a limitation.

Response:

We appreciate the reviewer's attention to this aspect. Not all participants were included in the study at T0. The missing samples were primarily due to participant unavailability or, in the case of T3, due to SARS-CoV-2 infection (which was the case for one aCD20-treated MS and NMOSD patient). Of note, due to the low number of infected patients, we could not draw any conclusions from the exclusive infection of aCD20-treated patients. A previous SARS-CoV-2 infection was an exclusion criterion during the screening process of the study. We have made sure to include the number of patients of each studied group in every figure. Furthermore, in the discussion section of the revised manuscript, we have acknowledged and described this limitation, as suggested. Furthermore, we provide information on dropouts in the methods section (page 17).

Is the methodology sound? Does the work meet the expected standards in your field?
The laboratory work is well performed and appears methodologically sound. There is no pre-specified protocol mentioned. The numbers are limited. The study in its current form is not powered nor designed appropriately to help inform us of the impact of the cellular changes identified.

Response:

As mentioned previously, we acknowledge that our study's small sample size may limit the strength of our conclusions regarding the impact of the identified cellular changes. However, our findings still offer relevant insights into multiple cell compartments involved in immune responses to DMTs and vaccinations. By focusing on lower-dimensional immune cell monitoring approaches, researchers can use the identified parameters to further explore the function, mechanisms, and actual impact of these cellular changes. Additionally, to gain a deeper understanding, it will be necessary to employ different approaches, such as vitro studies or animal models.

Is there enough detail provided in the methods for the work to be reproduced?
I cannot comment on all the techniques as I have not used them all.

REVIEWER COMMENTS

Reviewer #1 (expert in neurology, MS, NMOSD):

Thanks. I think this study is ready to publish

Reviewer #2 (expert in mass cytometry, single-cell analysis, SARS-CoV2):

Wang et al. have addressed my original comments and concerns. I have a few additional minor comments that should be addressed before acceptance.

Minor comments:

- Fig3G, right plot. Even though statistically there's a negative correlation with G12, it doesn't look biologically real. It may be "significant" because of leverage, so try removing the patient highest "Anti-S1 IgG level at T1-T3" value to see whether it's still significant. Consider removing
- Fig5-7 add what the % are of for the y-axis of the correlation plots
- Fig6B G13 correlation looks very weak. As above, it may just be leverage from the one patient with a very high % and low StimIndex

Reviewer #3 (expert in neurology, MS, NMOSD):

Thank-you again. Sorry I am still finding it difficult to follow. Fundamentally I cannot see a take home message. Are the granulocyte compartments increased or relatively increased? and is the cause therefore loss of other compartments rather than granulocyte compartments? is NMOSD different from MS? It may be helpful to look at the significant results and remove data on treatments that are not powered to give an answer. The are is potentially relevant but I would like to know what I should monitor?

Abstract

no results are presented to clarify that there are no significant differences between NMOSD and MS? if I am correct.

Introduction

This is improved. But has lax language at times that is not supported by the data or references.

page 3 'neuropathology and demyelination' makes no sense. Demyelination is a type of neuropathology, neuropathology is not really a specific term

page 3 "In NMOSD, disease-specific auto..." in reality it seems there is no difference here between MS and NMOSD if one supposes the null hypothesis is not rejected

page 4 not really sure that 'monitoring the myeloid compartment' can be recommended on the basis of these numbers and should be removed. Monitoring what?

I understand specific DMT but what do you mean by specific profile and risk factors these are undefined terms and fairly meaningless

Results

page 6: "It is important to note that the

results obtained from NMOSD patients in this study should be considered as observations rather than statistically significant findings, given the ... surely this means you should not draw conclusions from them.

page 7 bottom. showed a positive correlation do you mean that higher levels of b cell retention is associated with better antibody responses?

page 8. additionally... this sentence does not read well

page 8 "Our results demonstrated treatment-dependent increase" is this proportion due to loss of other cell types?

page 8 are positive correlations useful. I assume making more antibodies is good?

page 9 "positive correlation, $p=0.0741$ " not significant

Discussion

page 12 first paragraph the english needs revision

what is the significance of a positive or negative correlation. is it good or bad?

This would benefit from a summarising of the results presented.

the sentence "Regardless of the specific infection.." seems very general and is repeated in different forms

Increased proportion of granulocytes is this because other compartments have reduced or is it absolute?

page 13 "In our analyses, we characterized heterogeneity of granulocytes and detected proportional changes .." is the change in proportion the cause or the loss of CD20 the reason?

"These changes may consequently affect risk of infection and/or responses to vaccination." is repeat of a statement a few lines above.

The conclusion seems to state the issues. Do CD20's affect the innate immune system that either supports or inhibits B and T cells reactivity to antigens. I would hope this would have been clarified by this work at to some extent.

REVIEWER COMMENTS

Reviewer #1 (expert in neurology, MS, NMOSD):

Thanks. I think this study is ready to publish.

Response:

We appreciate the reviewer's positive feedback.

Reviewer #2 (expert in mass cytometry, single-cell analysis, SARS-CoV2):

Wang et al. have addressed my original comments and concerns. I have a few additional minor comments that should be addressed before acceptance.

Response:

We would like to thank the reviewer once again for the valuable suggestions and comments, which have significantly contributed to the improvement of our manuscript.

Major comments:

• *Fig3G, right plot. Even though statistically there's a negative correlation with G12, it doesn't look biologically real. It may be "significant" because of leverage, so try removing the patient highest "Anti-S1 IgG level at T1-T3" value to see whether it's still significant. Consider removing.*

Response:

After removing the outlier (i.e., highest anti-S1 IgG value) and retesting the correlation, we were unable to confirm the significance of this cluster (G12); it is not statistically significant. Therefore, as advised, we have excluded the correlation with G12 from Figure 3 and have revised the text accordingly (see page 9).

• *Fig5-7 add what the % are of for the y-axis of the correlation plots.*

Response:

We apologize for the missing information. We have revised Figures 5-7 by including the percentage of all T cells/M-N cells/granulocytes on the y-axis of these figures.

• *Fig6B G13 correlation looks very weak. As above, it may just be leverage from the one patient with a very high % and low StimIndex.*

Response:

Following the suggestion, we have re-analyzed the data after excluding the mentioned outlier, and we can now confirm the significance of this correlation ($p=0.0415$). The figure legend has been updated to reflect this change.

Reviewer #3 (expert in neurology, MS, NMOSD):

Thank-you again. Sorry I am still finding it difficult to follow. Fundamentally I cannot see a take home message.

Response:

We thank the reviewer for this comment. However, it would be helpful if the reviewer could pinpoint specific parts of the manuscript that are unclear or challenging to follow.

In summary, our study yields the following key findings:

1. It highlights the impact of DMTs on the immune cell composition in patients with NMOSD and MS, both before and after initial as well as subsequent antigen encounters. Our results shed light on the varied nature of these changes across different treatments and diseases.
2. We uncover intriguing alterations within the innate immune cell compartment, including CD64⁺ granulocytes, CXCR4⁺ granulocytes, and CD8^{hi} NK cells, alongside the lymphocyte compartment.
3. These findings offer novel perspectives on the complex relationship between DMTs and immune cell composition. Moreover, our study reveals potential associations between specific alterations in innate immune cells and vaccine responses. This highlights the relevance of innate immune cells in the context of vaccination and adds depth to our understanding of the immunological landscape in DMT-treated patients with neuroimmunological diseases.

However, as we have acknowledged in the point-by-point (PbP) responses during the first revision, we are aware of the limitations of our study, which include a small sample size and the lack of functional investigations. We concur with the reviewer's viewpoint that further research is imperative to elucidate how these modifications in immune cell composition may affect infection rates and susceptibility, and disease severity. To delve into this aspect, the development of an appropriate experimental model is a prerequisite.

- Are the granulocyte compartments increased or relatively increased? and is the cause therefore loss of other compartments rather than granulocyte compartments?

Response:

As we have previously discussed in the PbP responses of the first revision, interpreting the results solely based on changes in proportions can be misleading. It is important to keep in mind that an "increased" cluster may not necessarily indicate an actual increase in cell numbers but rather a relative prominence due to a decrease in other populations. This effect becomes particularly evident when comparing different cell populations, as shown in Figure 1, where B and T cells are depleted, causing myeloid cells to appear dominant. In this current study, we specifically compare the cellular composition within a single population, such as granulocytes (as shown in Figure 3). In these cases, the proportion of sub-populations is calculated based on the total number of cells within that specific population (e.g., the proportion (%) of a sub-population G1 within total granulocytes). Consequently, the proportion of sub-populations within each cell population is not affected by other populations.

To address this point and ensure clarity, we have revised the manuscript accordingly (during the first revision), providing a clear explanation of the approach and its implications.

is NMOSD different from MS? It may be helpful to look at the significant results and remove data on treatments that are not powered to give an answer. The are is potentially relevant but I would like to know what I should monitor?

Response:

Our results have demonstrated both similarities and differences between MS and NMOSD, particularly in changes within the myeloid compartment. For instance, in Figure 1, we found that compared to untreated MS patients, granulocyte (C1) proportions increased in MS patients treated with aCD20 antibodies, FTY, and IFN β . In contrast, aCD20-treated NMOSD patients exhibited lower proportions of granulocyte (C1) than their untreated counterparts.

Similar differences were also evident in monocyte C14 (Figure 1) and granulocyte subclusters (Figure 3).

Furthermore, we detected that the correlations between changes in the myeloid cell compartment and immune responses after vaccination differed between diseases and types of treatments. For example, as shown in Figure 6, CD64⁺ granulocytes were positively correlated with S-I-specific CD4⁺ T cell reactivity in aCD20-NMOSD patients but negatively correlated in FTY-treated MS patients. The HLA-DR⁺ granulocyte G7 cluster was positively linked to S-I-specific CD4⁺ T cell reactivity in untreated MS patients but negatively associated in aCD20-treated MS patients. We have also taken the reviewer's suggestion into account and removed all plots that depict statistically non-significant results from the manuscript to ensure clarity.

In response to the reviewer's question regarding what should be monitored, our study suggests that characterizing sub-populations of granulocytes may provide valuable insights into evaluating immune responses in MS and NMOSD patients. However, before implementing these suggestions in routine clinical practice, further evaluation of our results in a larger cohort is necessary. Nevertheless, our findings are significant in defining a small set of cell type candidates for immune profiling in large-scale studies using a reduced-dimensional technology such as flow cytometry, which offers a cost-effective and accessible alternative to single-cell mass cytometry.

Abstract

no results are presented to clarify that there are no significant differences between NMOSD and MS? if I am correct.

Response:

Regarding the abstract, we refrained from providing a detailed description of the results due to space constraints.

Introduction

This is improved. But has lax language at times that is not supported by the data or references.

page 3 'neuropathology and demyelination' makes no sense. Demyelination is a type of neuropathology, neuropathology is not really a specific term.

Response:

We thank the reviewer for this correction. We have removed the term "neuropathology" from the introduction (page 3).

page 3 "In NMOSD, disease-specific auto..." in reality it seems there is no difference here between MS and NMOSD if one supposes the null hypothesis is not rejected.

Response:

While MS and NMOSD share some similarities in terms of neuropathology, they also exhibit notable differences. In NMOSD, autoreactive IgG1 targets the AQP4 water channel protein on astrocytes, leading to astrocytopathy, secondary demyelination, and neuron loss. In contrast, MS is characterized by demyelination as a key feature of lesions, with astrocytes and axons largely preserved (Lopez JA, Denkova M, et al. Clin Transl Immunology. 2021; Yokote H, Mizusawa H. Neural Regen Res. 2016). In our study, we focused on characterizing the immune profiles of both diseases under different conditions, revealing both similarities and differences, as mentioned above.

page 4 not really sure that 'monitoring the myeloid compartment' can be recommended on the basis of these numbers and should be removed. Monitoring what?

Response:

We acknowledge the reviewer's valid concern regarding the small sample size of our study, and we have duly addressed this limitation in our manuscript. However, our study has identified granulocyte populations that undergo proportion changes in response to DMT treatments. We have further investigated and characterized these granulocyte populations in relation to cellular and humoral responses following vaccination. Our findings shed light on the heterogeneity and important roles of granulocytes, which are often considered homogenous and underestimated, in immune responses to vaccination. This involvement of granulocytes, along with other myeloid cells, becomes more prominent when the T and B cell compartments are compromised, as observed under DMT treatment. Of note, neutropenia, a rare complication associated with anti-CD20 antibody therapies used in MS and other conditions, can increase the risk of infection (Rauniyar et al. 2022 (PMID: 35079395); Cohen et al. 2019 (PMID: 30635476); Malpica Castillo et al. 2020 (PMID: 32017201)). Thus, beyond their involvement in infections, granulocytes may also play a role in responding to various vaccines. Monitoring granulocytes and their heterogeneity alongside lymphocyte populations may aid physicians in better assessing a patient's immune status.

I understand specific DMT but what do you mean by specific profile and risk factors these are undefined terms and fairly meaningless.

Response:

We appreciate the reviewer's comment and apologize for any lack of clarity in our statement.

By "patient's disease profile", we refer to various aspects of the underlying neuroimmunological disease, including disease subtype, duration, and severity. To specify, we have revised the corresponding section to "the patient's neuroimmunological disease phenotype" and have provided examples. Similarly, when we refer to "risk factors", we mean elements that could influence a patient's susceptibility to adverse outcomes, treatment responses, or vaccine effectiveness, such as aging, lifestyle, and comorbidities. To clarify, we have included exemplary risk factors in the text.

Results

page 6: "It is important to note that the results obtained from NMOSD patients in this study should be considered as observations rather than statistically significant findings, given the ... surely this means you should not draw conclusions from them.

Response:

In our manuscript, we have addressed the limitations posed by our small sample size and have been cautious in drawing conclusions. As stated, the results obtained from NMOSD patients should be considered as observations. Based on these observations, we have offered insights into potential occurrences in these patients under varying conditions while avoiding any overstated conclusions.

In this revised manuscript, we have taken steps to explicitly emphasize the need for larger-scale studies to confirm and validate these initial observations (page 15).

page 7 bottom. showed a positive correlation do you mean that higher levels of b cell retention is associated with better antibody responses?

Response:

Our study identified a positive correlation between the proportion of several CD20⁺ B cell sub-clusters, which are affected by treatment, and the levels of anti-S1 IgG antibodies following vaccination. While this correlation suggests an association between B cells and

antibody response, this relationship is multifaceted and may not necessarily represent a straightforward cause-and-effect scenario. It's important to recognize that B cell sub-clusters play diverse roles within the immune response, and their presence may indicate a complex interplay between different processes. While we do observe a positive correlation between certain B cell subpopulations and antibody levels, it would be premature to conclude a direct causal relationship based solely on this correlation. Several factors, including the differentiation stage of B cells, the inflammatory microenvironment, and the functionality of specific B cell subsets, could contribute to this observed link. In summary, the observed correlation suggests an intricate interaction between B cell sub-clusters and antibody responses, but additional functional assays are necessary to fully decipher the underlying mechanisms and potential cause-and-effect dynamics.

page 8. additionally... this sentence does not read well

Response:

We thank the reviewer for this comment. We have revised the sentence for better readability:

page 8 "Our results demonstrated treatment-dependent increase" is this proportion due to loss of other cell types?

Response:

In this analysis, we focus specifically on comparing the cellular composition within granulocyte subpopulations. In these cases, the proportion of subpopulations is calculated based on the total number of granulocytes, thus ensuring that any increase in proportion is not masked by the loss of other cell types.

page 8 are positive correlations useful. I assume making more antibodies is good?

Response:

We thank the reviewer for this valuable feedback and question. The positive correlation we identified suggests that higher levels of HLA-DR^{low/dim} granulocyte sub-clusters are associated with elevated anti-S1 IgG antibody levels. While it's tempting to assume that generating more antibodies is unequivocally beneficial, it's essential to consider the context and nuances of immune responses.

Producing antibodies is a fundamental aspect of the immune response, particularly in response to vaccinations and pathogen exposure. Antibodies, such as anti-S1 IgG antibodies, play a pivotal role in neutralizing pathogens and preventing infections. Therefore, higher antibody levels could indicate an enhanced immune reaction against the pathogen of interest, which, from a protective standpoint, is generally considered advantageous. However, it's important to recognize that immune responses are intricate and multifaceted. An increased antibody response may also reflect a more robust immune reaction, potentially involving a broader array of immune cells and mechanisms. In some cases, an excessively intense immune response could lead to unintended consequences, such as immune-related adverse events. So, while it's reasonable to associate increased antibody levels with enhanced immune responses, the dynamic nature of immune reactions underscores the need for balanced immune regulation and the sustainability of functional antibody responses over the long term. Our study provides a foundation for further exploration of these correlations and a deeper understanding of the intricate interplay between innate immune cells and antibody production.

page 9 "positive correlation, p=0.0741" not significant

Response:

We appreciate the reviewer's comment. We have revised the sentences and removed all results that were not statistically significant (page 9).

Discussion

page 12 first paragraph the english needs revision

Response:

We thank the reviewer for this suggestion. In response to the feedback, we have carefully reviewed the entire paragraph. Specifically, we have revised the first sentence by dividing it into two sentences to enhance readability.

*what is the significance of a positive or negative correlation. is it good or bad?
This would benefit from a summarising of the results presented.*

Response:

We appreciate the reviewer's query regarding the significance of positive or negative correlations in our study and whether they denote favorable or unfavorable outcomes. In our study, the observed correlations hold valuable implications, as they highlight the intricate interactions between innate immune cell sub-populations and the broader immune response to vaccination. However, it's important to avoid oversimplification by categorizing these correlations as solely "good" or "bad." The immune response is highly complex and context-dependent. An increased distribution of specific innate immune cell sub-populations showing positive correlations with cellular and humoral responses might indicate a coordinated and robust immune reaction. This could suggest a favorable scenario where the immune system is effectively responding to vaccination. Conversely, negative correlations could suggest regulatory mechanisms that maintain immune balance. While a decrease in certain innate immune cell sub-populations might correlate with enhanced antibody or cellular responses, it could also imply intricate immunomodulatory processes at play.

In essence, the significance of these correlations lies in their ability to unveil dynamic immune interactions. Rather than assessing them as universally positive or negative, their value lies in contributing to a comprehensive understanding of immune responses to vaccination and potential infection risk. These correlations shed light on the complex orchestration of the immune system's components, enhancing our insight into the mechanisms governing immune reactions.

the sentence "Regardless of the specific infection.." seems very general and is repeated in different forms

Response:

The interpretation of this particular sentence, "Regardless of the specific infection or vaccine, antigens/pathogens stimulate cellular and humoral immune responses through similar mechanisms.", depends on the context. In essence, this sentence conveys that, in general, both infections and vaccines stimulate the immune system through similar mechanisms.

Increased proportion of granulocytes is this because other compartments have reduced or is it absolute?

Response:

As mentioned earlier, in this case, an "increased" proportion of granulocytes signifies relative prominence due to depletion in other populations.

However, it's important to note that changes in the proportion of granulocyte subpopulations were calculated based on the total number of cells within the granulocyte population, and thus, these changes were not affected by other cell types.

page 13 "In our analyses, we characterized heterogeneity of granulocytes and detected proportional changes .." is the change in proportion the cause or the loss of CD20 the reason?

Response:

As mentioned earlier, in this case, the calculation of the proportion of granulocyte subpopulations remains unaffected by other cell types.

"These changes may consequently affect risk of infection and/or responses to vaccination." is repeat of a statement a few lines above.

Response:

We have carefully reviewed these two similar sentences and believe they describe slightly different situations. Therefore, we would like to retain both of these sentences as they appear in the first revision.

The conclusion seems to state the issues. Do CD20's affect the innate immune system that either supports or inhibits B and T cells reactivity to antigens. I would hope this would have been clarified by this work at to some extent.

Response:

In our study, we have identified specific subpopulations of immune cells within the myeloid compartments that are affected by DMTs and exhibit significant correlations with both cellular and humoral responses following exposure to antigens, as exemplified by the SARS-CoV-2 mRNA vaccination. This observation highlights that DMTs, including aCD20 antibodies, have the potential to impact the innate immune system. However, further *in vitro* or *in vivo* studies, potentially using animal models, are required to precisely investigate the functional changes in B and T cells under stimulated conditions. Our findings provide a set of promising candidates within the innate immune system.

REVIEWERS' COMMENTS

Reviewer #2 (expert in mass cytometry, single-cell analysis, SARS-CoV2):

Wang et al. have addressed all my comments and concerns. I recommend the paper for publication.

Reviewer #3 (expert in neurology, MS, NMOSD):

Absent.